



# Azimuth-, angle- and frequency-dependent seismic velocities of cracked rocks due to squirt flow

Yury Alkhimenkov[1,2], Eva Caspari[3], Simon Lissa[1], and Beatriz Quintal[1,2]

[1]Institute of Earth Sciences, University of Lausanne, Lausanne, Switzerland
[2]Swiss Geocomputing Centre, University of Lausanne, Lausanne, Switzerland
[3]Chair of Applied Geophysics, Montanuniversity Leoben, Leoben, Austria

**Correspondence:** Yury Alkhimenkov (yury.alkhimenkov@unil.ch)

**Abstract.** Understanding the properties of cracked rocks is of great importance in scenarios involving $CO_2$ geological sequestration, nuclear waste disposal, geothermal energy, and hydrocarbon exploration and production. Developing non-invasive detecting and monitoring methods for such geological formations is crucial. Many studies show that seismic waves exhibit strong dispersion and attenuation across a broad frequency range due to fluid flow at the pore scale known as squirt flow. Nev-
ertheless, how and to what extent squirt flow affects seismic waves is still a matter of investigation. To fully understand its angle- and frequency-dependent behavior for specific geometries appropriate numerical simulations are needed. We perform a three-dimensional numerical study of the fluid-solid deformation at the pore scale based on coupled Lamé-Navier and Navier-Stokes linear quasistatic equations. We show that seismic wave velocities exhibit strong azimuth-, angle- and frequency-dependent behavior due to squirt flow between interconnected cracks. We show that the overall anisotropy of a medium mainly increases
due to squirt flow but in some specific planes the anisotropy can locally decrease. We analyze the Thomsen-type anisotropic parameters and adopt another scalar parameter which can be used to measure the anisotropy strength of a model with any elastic symmetry. This work significantly clarifies the impact of squirt flow on seismic wave anisotropy in three dimensions and can potentially be used to improve the geophysical monitoring and surveying of fluid-filled cracked porous zones in the subsurface.

# 1   Introduction

Cracked rocks have been under intensive studies during the last decades since they play a crucial role in $CO_2$ geological sequestration, nuclear waste disposal, geothermal energy and hydrocarbon exploration and production. Cracks and grain-scale discontinuities are the key rock parameters which control effective elastic and hydraulic properties of such rocks. Many studies show that seismic waves exhibit significant dispersion and attenuation in cracked porous rocks due to pore-scale fluid flow
(O'Connell and Budiansky, 1977; Dvorkin et al., 1995; Gurevich et al., 2010; Müller et al., 2010). Furthermore, cracks cause significant seismic wave anisotropy (Schoenberg and Sayers, 1995; Sayers and Kachanov, 1995; Sayers, 2002; Chapman, 2003; Maultzsch et al., 2003; Tsvankin and Grechka, 2011). Thus, seismic methods can be used to detect and characterize cracked zones and may be useful to predict crack density, their preferred orientation and interconnectivity of cracks.





One can define fractures as discontinuities at the mesoscopic scale and cracks as discontinuities at the pore scale. Fluid flow
due to a passing wave may happen at different scales: at the wavelength scale, at the mesoscopic scale and at the pore scale
(Müller et al., 2010). Biot's theory (Biot, 1962) describes the so-called global flow at the wavelength scale but its overall effect
on a passing wave at seismic frequencies is usually much smaller than that of fluid flow at the mesoscopic and pore scales
(Pride et al., 2004). Mesoscopic scale is that much larger than the pore-scale but smaller than the wavelength. At this scale,
studies are performed in the framework of Biot theory, assuming heterogeneous rock properties. There are several analytical
and numerical studies on the effect of wave-induced fluid flow between mesoscopic fractures and a porous rock background
and between interconnected fractures using the Biot's equations (Brajanovski et al., 2005; Rubino et al., 2013; Quintal et al.,
2014; Masson and Pride, 2014; Grab et al., 2017; Hunziker et al., 2018; Caspari et al., 2019) as well as on the comparison
between the numerical and analytical results (Guo et al., 2017, 2018). Experimental studies of synthetic rock samples showed
the impact of fluid-saturated fractures on seismic velocities (Amalokwu et al., 2016; Tillotson et al., 2012, 2014). The resulting
frequency-dependent anisotropy was analysed by Carcione et al. (2013); Rubino et al. (2017); Barbosa et al. (2017). The last
two also considered fracture-to-fracture flow, in addition to fracture-to-background flow.

At the pore scale, a passing wave induces fluid pressure gradients which occur between interconnected cracks, as well as,
between cracks and stiffer pores. Such pressure gradients force fluid to move between different cracks and pores until the
pore pressure equilibrates throughout the connected pore space. This phenomenon, known as squirt flow causes strong energy
dissipation due to the viscosity of the fluid and the associated viscous friction. Several experimental studies confirmed the
importance of squirt flow at different frequency ranges (Mayr and Burkhardt, 2006; Best et al., 2007; Adelinet et al., 2010;
Mikhaltsevitch et al., 2015; Pimienta et al., 2015; Subramaniyan et al., 2015; Chapman et al., 2019). There are several analytical
solutions for squirt flow (O'Connell and Budiansky, 1977; Dvorkin et al., 1995; Chapman et al., 2002; Gurevich et al., 2010)
which are based on simplified pore geometries and many physical assumptions.
Dispersion and attenuation caused by squirt flow can be simulated numerically by solving the coupled fluid-solid deformation
at the pore scale using Lame-Navier and Navier-Stokes equations with appropriate boundary conditions and, then, calculating
effective frequency-dependent viscoelastic properties. During the last decades, many studies used numerical methods to solve
mechanical problems (Andrä et al., 2013a, b; Saxena and Mavko, 2016). Recently, some numerical studies appeared in the geo-
physical literature aiming to solve the coupled fluid-solid deformation and, hence, studying dispersion and attenuation caused
by squirt flow (Zhang et al., 2010; Zhang and Toksöz, 2012; Quintal et al., 2016, 2019; Das et al., 2018; Alkhimenkov et al.,
2019). Das et al. (2018) numerically simulated a fully coupled fluid-solid interaction at the pore scale for digital rock sam-
ples. They modeled the pore fluids as Newtonian fluids using the Navier-Stokes equation with appropriate coupling between
both the solid and liquid phases, accounting for inertial effects. Quintal et al. (2016, 2019) simplified the Navier-Stokes equa-
tions by assuming compressible Stokes flow, neglecting inertial term and, hence, using the linearized quasistatic Navier-Stokes
equation.

We numerically simulate squirt flow in three dimensions and calculate frequency-dependent effective stiffness moduli using
the finite-element method to solve the quasi-static Lamé-Navier equations coupled to the linearized quasi-static Navier-Stokes
equations (Quintal et al., 2016, 2019; Alkhimenkov et al., 2019). We apply an oscillatory deformation to certain boundaries



of the numerical model, and, assuming that the wavelength is much larger than the size of individual cracks, we calculate the volume average stress and strain fields and the resulting effective stiffness moduli. Then, we calculate the associated azimuth-, angle- and frequency-dependent seismic velocities by solving the Christoffel equation. The main goal of this study is to analyse seismic anisotropy due to squirt flow in three dimensions since the previous numerical studies of seismic anisotropy were performed only in two dimensions and in the framework of Biot's theory (Rubino et al., 2017; Barbosa et al., 2017).

This paper is organized as follows. First, we briefly describe the numerical methodology. Then, we describe the numerical model and show the numerical results — frequency-dependent effective stiffness moduli. After, by solving the Christoffel equation, we evaluate the angle-, azimuth- and frequency dependent velocities of the model. Lastly, we quantify the anisotropy strength of the models analyzing the conventional Thomson-type anisotropy parameters and also by adopting another scalar parameter.

## 2  Numerical methodology

We consider that at the pore scale, a rock is composed by a solid material (grains) and a fluid-saturated pore space (cracks). The numerical methodology is described by Quintal et al. (2016, 2019) and Alkhimenkov et al. (2019) and here we briefly outline the main equations. The solid phase is described as a linear isotropic elastic material for which the conservation of momentum is (e.g., Landau and Lifshitz (1959b) and Nemat-Nasser and Hori (2013))

$$\nabla \cdot \boldsymbol{\sigma} = 0, \tag{1}$$

where "$\nabla \cdot$" denotes the divergence operator acting on the stress tensor $\boldsymbol{\sigma}$. The stress-strain relation for an elastic material can be written as

$$\boldsymbol{\sigma} = (K - \frac{2}{3}\mu)\mathbf{tr}\left(\frac{1}{2}\left((\nabla \otimes \mathbf{u}) + (\nabla \otimes \mathbf{u})^{\mathrm{T}}\right)\right)\mathbf{I}_2 + 2\mu\left(\frac{1}{2}\left((\nabla \otimes \mathbf{u}) + (\nabla \otimes \mathbf{u})^{\mathrm{T}}\right)\right) \tag{2}$$

where $\mathbf{I}_2$ is the second order identity tensor, $\mathbf{tr}$ is the trace operator, "$\otimes$" defines the tensor product, the superscript "$\mathrm{T}$" corresponds to the transpose operator, $K$ and $\mu$ are the bulk and shear moduli.

The fluid phase is described by the quasi-static linearised compressible Navier-Stokes momentum equation (Landau and Lifshitz, 1959a):

$$-\nabla p + \eta \nabla^2 v + \frac{1}{3}\eta \nabla \left(\nabla \cdot v\right) = 0, \tag{3}$$

where $v$ is the particle velocity, $p$ is the fluid pressure and $\eta$ is the shear viscosity. Equation (3) is valid for the laminar flow of a Newtonian fluid. In the finite element numerical solver, equations (2)-(3) are combined in the space-frequency domain so the solid and fluid displacements are described by the same variable and, thus, naturally coupled at the boundaries between subdomains (Quintal et al., 2016, 2019). In this simulation, the energy dissipation is caused only by fluid pressure diffusion, since inertial terms are neglected.

The whole spatial domain is discretized using an unstructured mesh with tetrahedral elements. A direct PARDISO solver (Schenk and Gärtner, 2004) is used for solving the linear system of equations. Direct relaxation tests are performed to compute





all components of the stiffness matrix (in Voigt notation) $c_{ij}$. The boundary conditions are applied to the external walls of the model. The initial conditions for displacements are set to zero. The resulting stress and strains are averaged over the spatial domain for each frequency. Then, the complex valued $c_{ii}(\omega)$ components (diagonal) are calculated for each frequency (in Voigt notation, no index summation):

$$c_{ii}(\omega) = \frac{\langle \sigma_i(\omega) \rangle}{\langle \epsilon_i(\omega) \rangle},$$ (4)

where $\langle \cdot \rangle$ represents the volume averaging over the sample volume.

For calculating the P-wave modulus ($ii = 11, 22, 33$), a harmonic displacement on the $i$ direction is applied perpendicularlly to a wall of the model. At the other walls of the model, the normal component of the displacement is set to zero. For calculating shear components of the stiffness matrix ($ii = 44, 55, 66$), the boundary conditions applied are those of a simple shear test. A detailed description of the boundary conditions is given in Alkhimenkov et al. (2019). For the $c_{12}(\omega)$, $c_{13}(\omega)$ and $c_{23}(\omega)$ components (off-diagonal), mixed direct tests are needed, and the corresponding boundary conditions are given in Appendix A. The corresponding inverse quality factor is (O'connell and Budiansky, 1978)

$$\frac{1}{Q_{ij}(\omega)} = \frac{\text{Im}\{c_{ij}(\omega)\}}{\text{Re}\{c_{ij}(\omega)\}}.$$ (5)

## 3 Numerical model

Two 3D numerical models are constructed, which consist of a pore space embedded into an elastic solid grain material (Figure 1). The solid grain material is represented by a cuboid whose size is $(0.24 \times 0.24 \times 0.24)$ m$^3$. The pore space consists of two perpendicular cracks represented by thin cylinders of $0.002$ m thickness, $0.1$ m radius (i.e., the aspect ratio is thickness divided by diameter — $0.01$) and fully saturated with a liquid. In the first model, the two cracks are disconnected, while, in the second model, the two cracks are connected (cross sections in Figure 1). The employed liquid properties are those of glycerol and the grain material has properties of quartz (Table 1).

A fine, regular mesh is used inside the crack to accurately account for dissipation, while in the grain material the mesh is coarser (Figure 2). The total number of elements is $3.3 \times 10^6$. The simulation is performed for $13$ different frequencies from $10^1$ to $10^7$ Hz. For each frequency, the solver uses approximately $0.95$ Terabyte of RAM memory.

One crack embedded into an isotropic background induces a transverse isotropy (5 independent components of the stiffness tensor e.g., Mavko et al. (2009)). If the crack is parallel to the $xy$-plane, then the symmetry is vertical and the medium exhibits vertical transverse isotropy — VTI symmetry. If the crack is parallel to the $xz$-plane, then the symmetry is horizontal and the medium exhibits horizontal transverse isotropy — HTI symmetry. If two cracks, perpendicular to each other are embedded into an isotropic material and the crack compliances are different, then the medium exhibits orthorhombic symmetry (9 independent components of the stiffness tensor). If the the crack compliances are the same, then the medium symmetry





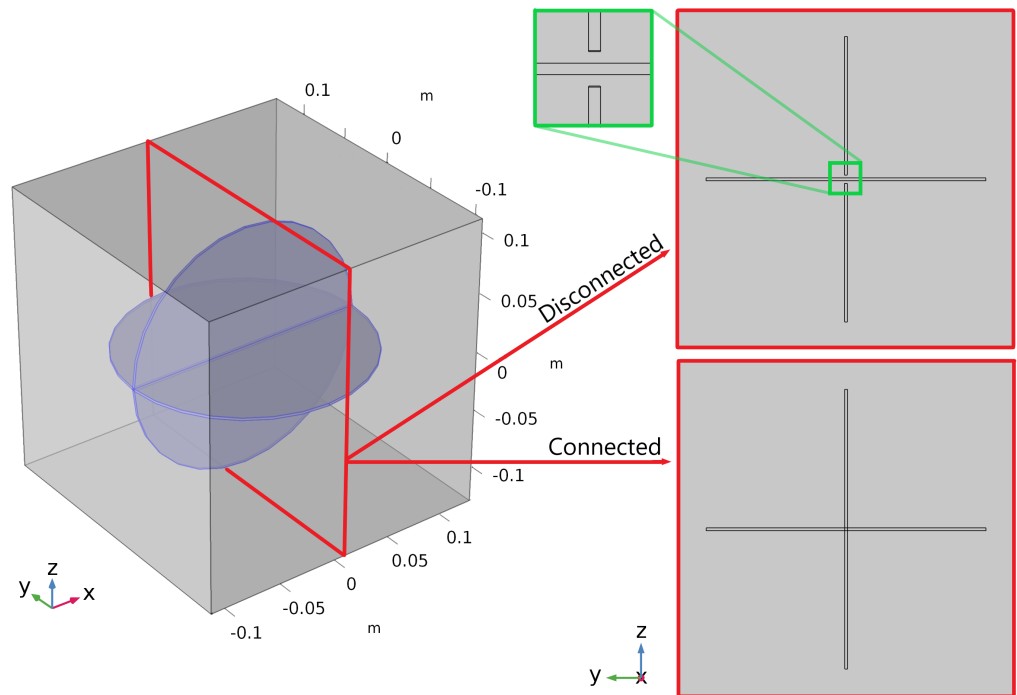

**Figure 1.** Sketch illustrating two flat cylinders representing two cracks. The blue region represents the pore space saturated with a fluid, the transparent gray area corresponds to the solid grain material. In the first model, the two cracks are disconnected as illustrated by the upper right cartoon. In the second model, the two cracks are connected as illustrated by the down right cartoon.

120    is tetragonal (6 independent components of the stiffness tensor); some authors attribute this geometry to a special case of orthorhombic symmetry (e.g., Bakulin et al. (2000b)), while tetragonal and orthorhombic symmetry classes are different. On the other hand, one can argue that an orthorhombic medium (created by two perpendicular sets of cracks) degenerates into a tetragonal medium if the cracks compliances are the same.

     The symmetry of the saturated numerical model with connected cracks is tetragonal (Figure 1) because the crack compliances

125    are the same. Thus, there are only six independent components of the stiffness tensor. We will see that the symmetry of the saturated numerical model with disconnected cracks is orthorhombic because one crack is stiffer than the other one due to its separation into two parts. But the difference between $c_{22}$ and $c_{33}$ stiffness components is less then $0.3\%$, thus the divergence from the tetragonal symmetry is negligible and, therefore, this model is considered as tetragonal as well.





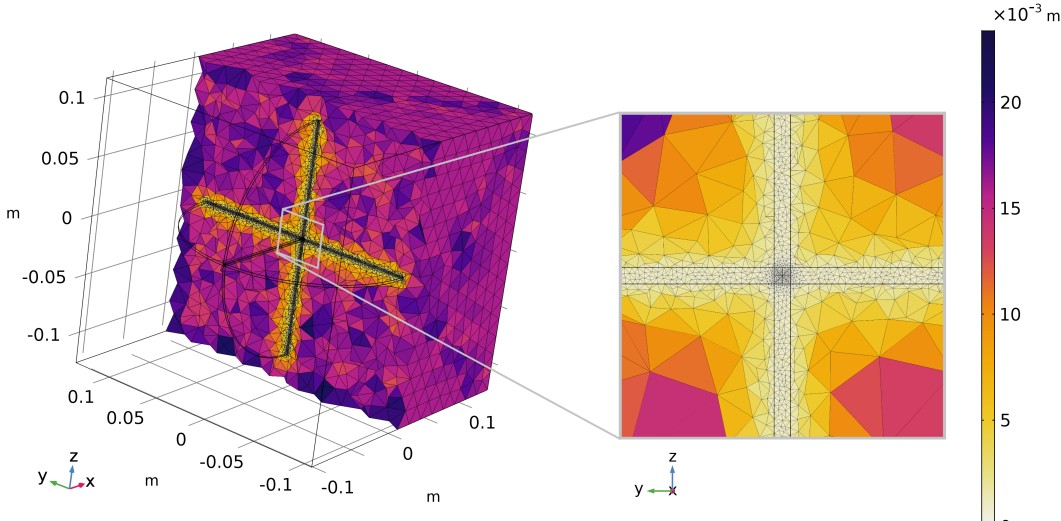

**Figure 2.** Sketch illustrating the element's size distribution for the model with connected cracks. Element's size in the crack is $5 \times 10^{-5} - 1 \times 10^{-3}$ m and in the surrounding grain material is $2.4 \times 10^{-3} - 1.6 \times 10^{-2}$ m. The element's size distribution for the model with disconnected cracks is the same.

**Table 1.** Material properties of the numerical model

| Material property | quartz | glycerol | air |
|---|---|---|---|
| Bulk modulus $K$ | 36 GPa | 4.3 GPa | $1.01 \times 10^{-4}$ GPa |
| Shear modulus $\mu$ | 44 GPa | — | — |
| Shear viscosity $\eta$ | — | 1.414 Pa·s | $1.695 \times 10^{-5}$ Pa·s |





## 4   Results

### 4.1   Dry stiffness moduli

Let's first consider the geometry shown in Figure 1 with a pore space filled with air (i.e. dry). We perform nine relaxation tests to calculate the full stiffness tensor for each of the two models with connected and disconnected cracks. The resulting effective stiffness moduli for the model with connected cracks are (in Voigt notation)

$$c_{ij}^{Con} = \begin{bmatrix} 93.53 & 4.65 & 4.65 & 0 & 0 & 0 \\ 4.65 & 63.91 & 5.46 & 0 & 0 & 0 \\ 4.65 & 5.46 & 63.91 & 0 & 0 & 0 \\ 0 & 0 & 0 & 31.62 & 0 & 0 \\ 0 & 0 & 0 & 0 & 35.16 & 0 \\ 0 & 0 & 0 & 0 & 0 & 35.16 \end{bmatrix} \text{(GPa)}. \tag{6}$$

For the model with disconnected cracks, the effective stiffness moduli are (in Voigt notation)

$$c_{ij}^{Dis} = \begin{bmatrix} 93.55 & 4.92 & 4.60 & 0 & 0 & 0 \\ 4.92 & 69.21 & 4.40 & 0 & 0 & 0 \\ 4.60 & 4.40 & 64.06 & 0 & 0 & 0 \\ 0 & 0 & 0 & 31.95 & 0 & 0 \\ 0 & 0 & 0 & 0 & 35.16 & 0 \\ 0 & 0 & 0 & 0 & 0 & 36.96 \end{bmatrix} \text{(GPa)}. \tag{7}$$

The effective stiffness moduli of the two models are different. Zero values are written if the value is below 0.0002 GPa (i.e. up to numerical precision). The $c_{ij}^{Con}$ stiffness matrix precisely belongs to the tetragonal symmetry class while the $c_{ij}^{Dis}$ stiffness matrix has all diagonal components different from each other, thus, it represents the orthorhombic symmetry class. The largest difference between $c_{ij}^{Con}$ and $c_{ij}^{Dis}$ is in the $c_{22}$ component, i.e., $\Delta c_{22} = c_{ij}^{Dis} - c_{ij}^{Con} = 5.3$ GPa. That is a significant difference and it is only due to the vertical crack separation.

There are two different features which must be clearly separated: 1) The effect of cracks intersections without changing the cracks geometry on the effective elastic properties. In this case, the crack intersection is achieved by changing the spatial position of the cracks. Grechka and Kachanov (2006) studied numerically the effect of cracks intersections without changing the crack geometry. They concluded that crack intersections have a very little impact on the effective elastic moduli. In this study, we formally consider only the model with connected cracks and we show that cracks can be accurately described by only two compliances which can be seen in equation (6). 2) The effect of the crack partition into two "halves" on the effective elastic properties. In this case, the partitioned crack has a long thin contact across the whole diameter (Figure 1). It is known





that the islands of contacts inside a crack significantly reduce crack compliance (a crack with contact areas is stiffer compared to the same crack but without contacts (Trofimov et al., 2017; Kachanov and Sevostianov, 2018; Markov et al., 2019; Lissa et al., 2019)). Comparing eq. (6) and (7), this study also shows that thin contact area significantly reduce crack compliance: the effective dry moduli of the model with disconnected cracks are much stiffer compared to the model with connected cracks.

An intuitive explanation is the following: if the cracks surface have not been changed by changing the spatial position of the cracks in the volume — the effect of crack intersections is negligible (Grechka and Kachanov, 2006); if the cracks surface have been changed, as we did in the present study by partitioning the vertical crack into two pieces (and introducing a thin additional contact area) — the effective elastic moduli would become much stiffer compared to the model where the cracks surface has not been changed.

## 4.2   Fluid pressure fields


Here and later on we deal only with a liquid-saturated pore space. The liquid has properties of glycerol (Table 1). A direct P-wave modulus test is performed to calculate dispersion and attenuation of the $c_{33}$ component (a harmonic displacement is applied to the top wall of the model in $z$-direction, while the normal component of the displacement is set to zero on all the other walls). Figure 3 shows snapshots of the fluid pressure $P_f$ in the cracks at three different frequencies, in the vertical middle

slice of the model (the $yz$ plane, red frame in Figure 1 (left)). For the model with connected cracks, at low frequencies, there is enough time for pressure equilibration between the cracks, thus, the pore pressure is uniform throughout the pore space (Figure 3, LF (connected)). This is called the relaxed state. At intermediate frequencies, there is a large pressure gradient in the cracks, which corresponds to the maximum attenuation due to squirt flow between cracks (Figure 3, Fc (connected)). At high frequencies, there is no time for fluid to move, hence, there is no fluid pressure equilibration between the vertical and horizontal

cracks (Figure 3, HF (connected)). This is called the unrelaxed state. Therefore, at high frequencies, the connected cracks behave as hydraulically isolated and the fluid highly stiffens the crack. In the model with disconnected cracks, the fluid pressure in the cracks is the same in all three regimes which corresponds to the unrelaxed state in the model with connected cracks. The unrelaxed state can be interpreted as the elastic limit because there is no fluid flow between the cracks and the effective properties of the two models (connected and disconnected cracks) are the same, as will be shown in the next subsection.

## 4.3   Dispersion and attenuation


### 4.3.1   Elastic moduli

Figure 4 shows the numerical results for the complex-valued frequency-dependent components of the stiffness matrix $c_{ij}(\omega)$ (in Voigt notation) for the models with connected and disconnected cracks filled with glycerol. In the model with connected cracks, the dispersion (the real part of the $c_{ij}$ component) and attenuation (eq. (5)) curves show strong frequency-dependent

behavior of the $c_{22}$, $c_{33}$ and $c_{23}$ components (Figure 4, (a) and (c)). The attenuation and dispersion of the $c_{22}$, $c_{33}$ components coincide because the geometrical properties of the two cracks are the same (Figure 4, (a)) and the model is symmetric. The $c_{11}$ component is non-dispersive and exhibit zero attenuation. The dispersion of the $c_{44}$, $c_{55}$ and $c_{66}$ components is negligible and





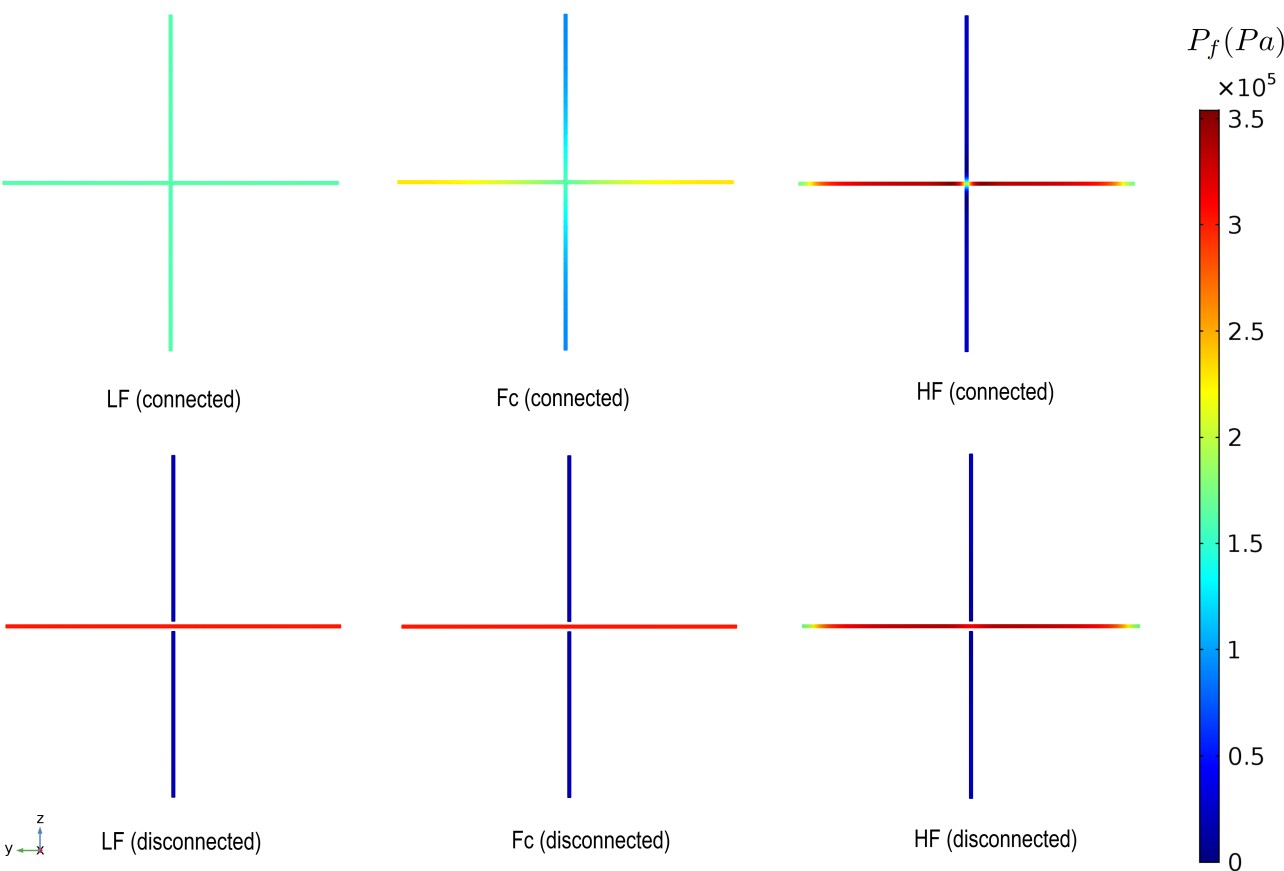

**Figure 3.** Snapshots of the fluid pressure $P_f$ in the cracks at three different frequencies: Lf - the low frequency (relaxed state), Fc - intermediate frequency (close to the characteristic frequency) and HF - the high frequency (unrelaxed state).

these components exhibit also negligible attenuation (Figure 4, (b)). The $c_{23}$ component exhibits strong negative dispersion and the attenuation peak is shifted towards high frequencies compared to that of the $c_{22}$, $c_{33}$ components. The $c_{12}$ and $c_{13}$
components are non-dispersive and exhibit zero attenuation.

In the model with disconnected cracks, all components of the stiffness tensor $c_{ij}(\omega)$ (Figure 4, (a-c)) are constant across the whole frequency range and exhibit zero attenuation. Furthermore, all components are equal to the high frequency values of the model with connected cracks. This is expected in the unrelaxed state because the connected cracks behave as hydraulically isolated with respect to fluid flow. A very small discrepancy between the two models at high frequencies is associated with the
vertical crack partition (two thin regions of pore space replaced with stiffer grain material).





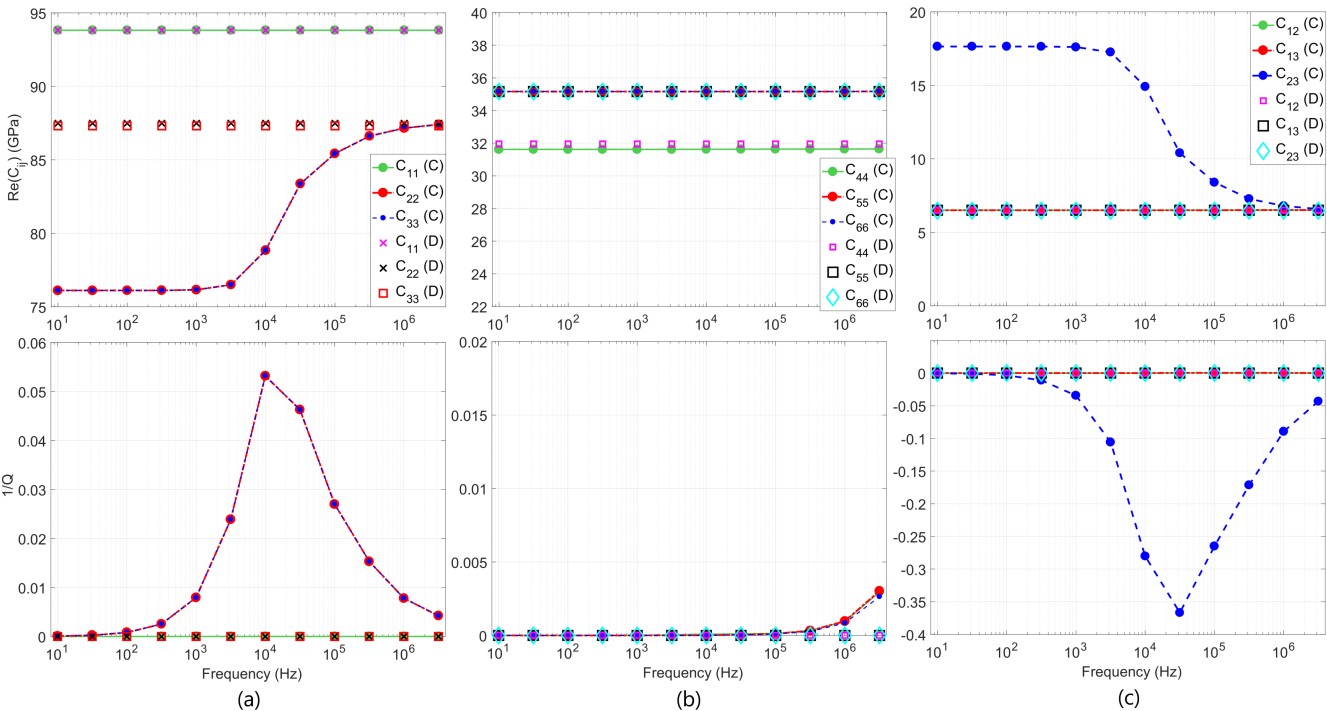

**Figure 4.** Numerical results for the connected (C) and disconnected (D) cracks models: (a-c) Real part of the $c_{ij}$ components versus frequency (upper plots), dimensionless attenuation of the $c_{ij}$ components versus frequency (lower plots). Each symbol corresponds to one numerical simulation and lines correspond to linear interpolation between discrete numerical results.

### 4.3.2 Seismic velocities

Figure 5 shows the P-wave (primary wave) phase velocity as a function of the phase angle of the numerical model with connected and disconnected cracks (Figure 1), where the zero phase angle corresponds to the vertical wave propagation (along $z$- axis). Considering the plane $YZ$, the P-wave velocity is the same for phase angles of $0$ and $90$ degrees, it changes with frequency only for phase angles between $0$ and $90$ degrees and is maximal in the high frequency limit at phase angle of $\theta = 90(\pm 90)$ degrees (Figure 5, left). Furthermore, in the high frequency limit the P-wave phase velocity coincides for the models with connected and disconnected cracks. As frequency decreases, the P-wave velocity decreases and at $10^4$ Hz the P-wave velocity is almost angle independent (yellow curve, Figure 5, left). It is interesting that this "local" isotropy corresponds to the maximum attenuation of the $c_{22}$ and $c_{33}$ components (Figure 4). As frequency further decreases, the P-wave velocity decreases and stays nearly unchanged for the frequencies below $10^{3.5}$ Hz. In the $XZ$-plane, the P-wave phase velocity is the same for the models with connected and disconnected cracks in the high frequency limit (Figure 5, right). For the model with connected cracks, as frequency decreases, the P-wave velocity decreases, reaching its minimum at low frequencies ($10^1$-$10^{3.5}$ Hz).





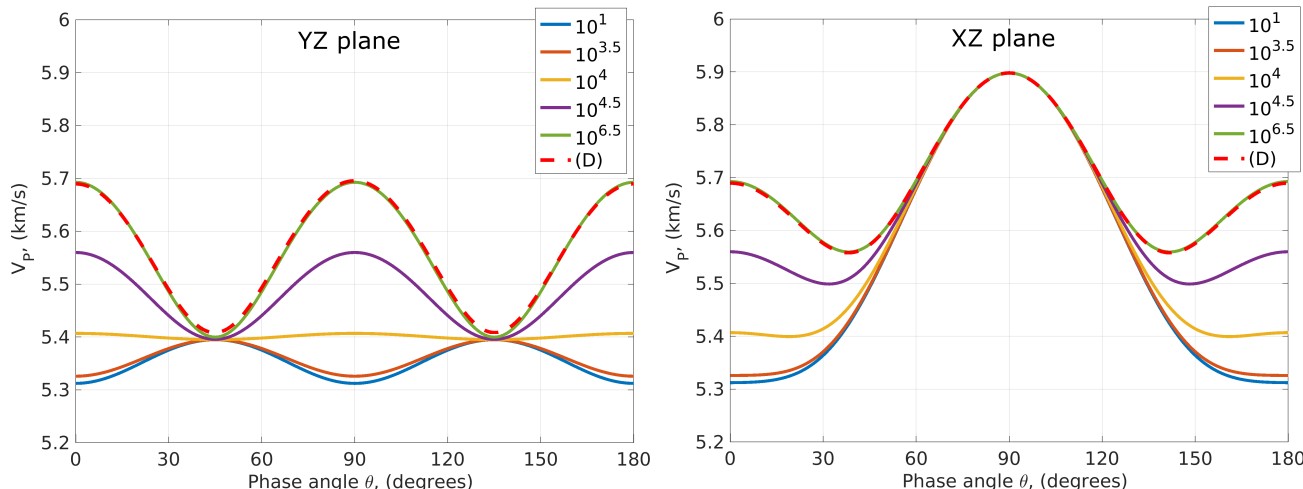

**Figure 5.** P-wave phase velocity versus phase angle in the $YZ$-plane (left) and the $XZ$-plane (right). Curves $10^1$-$10^{6.5}$ denote the frequency of the P-wave for the model with connected cracks. (D) — denotes the P-wave for the model with disconnected cracks.

Figures 6-7 show the quasi-shear (SV) and the pure shear (SH) phase velocities as functions of the phase angle of the numerical models with connected and disconnected cracks (Figure 1). The SV-wave velocity is strongly frequency-dependent in both the $XZ$- and $YZ$ planes. The SH-wave exhibits some frequency-dependent behavior in the $XZ$-plane and is angle- and frequency- independent in the $YZ$ plane. It is interesting that the SV-waves in two different planes have different velocities at $0$ and $90$ phase angles, which is due to their different wave polarization. The SV-wave in the $YZ$ plane has the same polarization as the SH-wave in the $XZ$ plane, their velocities are equivalent at the $0$ and $90$ phase angles. The same conclusion is valid for the SV-wave in the $XZ$ plane and the SH-wave in the $YZ$ plane.

Due to the symmetry of the model, the behaviors of the P-, SV-, SH-wave phase velocities in the $XZ$- and $XY$-planes are the same, thus the results in the $XY$-plane are not shown here.

### 4.4 Quantitative analysis of the frequency-dependent anisotropy

First, we quantify the Thomsen-type anisotropic parameters (Thomsen, 1986) for orthorhombic media (Tsvankin, 1997; Bakulin et al., 2000a). Then, we quantify the universal elastic anisotropy index (Ranganathan and Ostoja-Starzewski, 2008) and the two parameters which define the anisotropy strength in bulk and shear modes. All these anisotropy measures highlight different frequency-dependent features of the models. Our results shown in Figure 4 (frequency-dependent elastic moduli) are used as input to quantify these anisotropy measures.




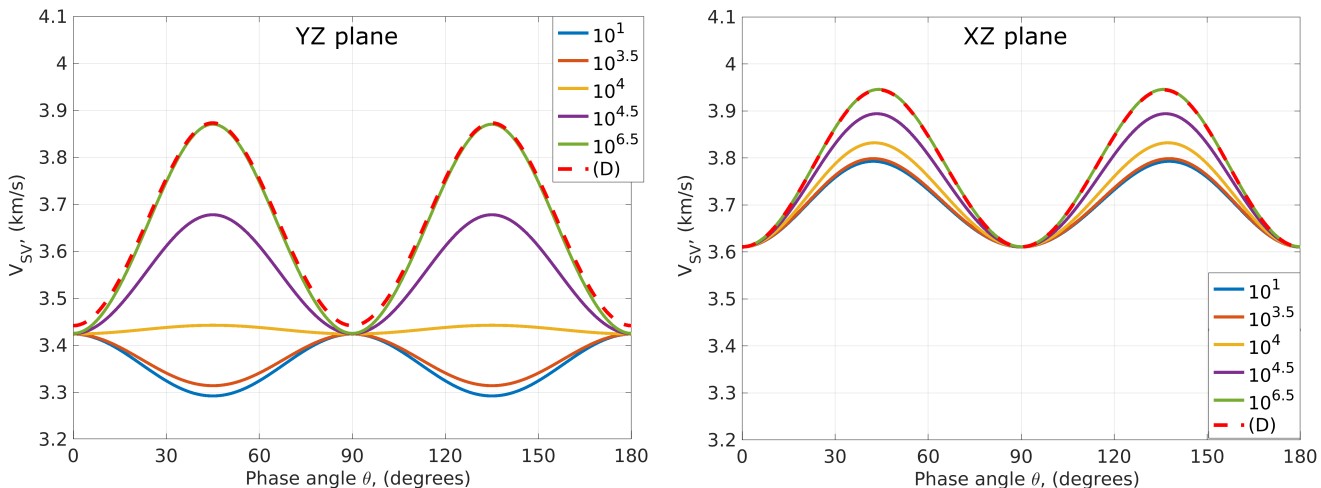

**Figure 6.** SV-wave phase velocity versus phase angle in the $YZ$-plane (left) and the $XZ$-plane (right). Curves $10^1$-$10^{6.5}$ denote the frequency of the P-wave for the model with connected cracks. (D) — denotes the P-wave for the model with disconnected cracks.

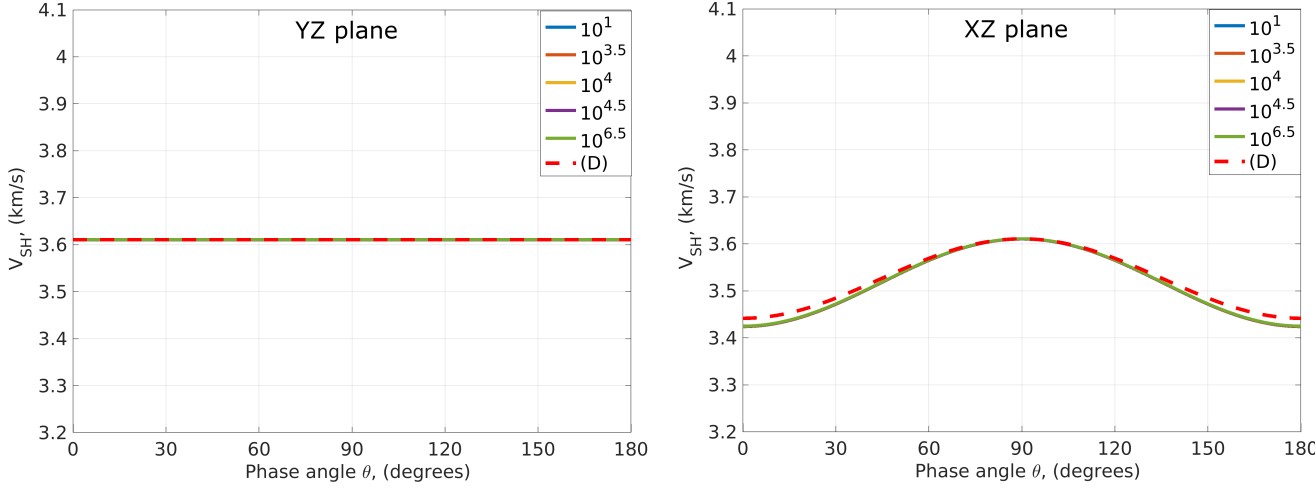

**Figure 7.** SH-wave phase velocity versus phase angle in the $YZ$-plane (left) and the $XZ$-plane (right). Curves $10^1$-$10^{6.5}$ denote the frequency of the P-wave for the model with connected cracks. (D) — denotes the P-wave for the model with disconnected cracks.





### 4.4.1 Thomsen-type parameters

Thomsen-type anisotropic parameters ($\epsilon$, $\delta$, $\gamma$) describe the P-wave anisotropy — $\epsilon$, the shape of the P-wave phase velocity at different phase angles — $\delta$ and the S-wave anisotropy — $\gamma$: each set of three parameters corresponds to one plane. Thus, for our model symmetry, there are two different planes — $YZ$ and $XZ$ (because the $XZ$ plane is equivalent to the $XY$ plane). In this study, we refer to Thomsen parameters $|\epsilon|,|\delta|,|\gamma| \in [0;0.1]$ as to weak elastic anisotropy ($|\cdot|$ corresponds to the absolute value), $|\epsilon|,|\delta|,|\gamma| \in [0.1;0.15]$ as to moderate elastic anisotropy and $|\epsilon|,|\delta|,|\gamma| \in [0.15;+\infty]$ as to strong elastic
anisotropy. The choice of these intervals is based on the divergence between the exact and approximate (by using Thomsen parameters) equations for the $P$-wave phase velocities in cracked media.

Figure 8 shows the Thomsen-type anisotropy parameters in the $YZ$ and $XZ$ planes (formulas are given in Appendix B). In the high frequency limit, all anisotropy parameters are the same for both models with connected and disconnected cracks. Furthermore, for the model with disconnected cracks, all anisotropy parameters are frequency independent because the stiffness
tensor is frequency independent. For the model with connected cracks, several anisotropy parameters are frequency dependent due to squirt flow.

In the $YZ$ plane, parameters $\epsilon^{YZ}$ and $\gamma^{YZ}$ are zero for both models. The parameter $\delta^{YZ}$ is frequency dependent and controls the shape of the $P$- wave phase velocity between 0 and 90 degrees. In the high frequency limit, $\delta^{YZ}$ exhibits the maximum negative value which correponds to strong elastic anisotropy. As frequency decreases, $\delta^{YZ}$ also decreases reaching zero value
around $10^4$ Hz and, then, $\delta^{YZ}$ increases reaching its positive maximum at low frequencies which corresponds to weak elastic anisotropy; the positive maximum is approximately $1/3$ of the absolute value of its negative maximum. It is interesting, that $\delta^{YZ}$ changes sign from negative to positive which is indeed observed in the P-wave velocity behavior (Figure 5, left) as P-wave velocity changes polarity with frequency. This was also observed by Barbosa et al. (2017) in the framework of Biot's theory. This polarity change has a fully mechanical nature. In the high frequency limit, cracks behave as hydraulically isolated and
fluid highly stiffens the normal compliance of the cracks (not tangential). As frequency decreases, fluid started to flow from more compliant to stiffer cracks as a response to the applied displacement boundary condition. $\delta^{(yz)} = 0$ corresponds to zero anisotropy; the numerator of $\delta^{YZ}$ is $[c_{23}(\omega) + c_{44}(\omega)]^2 - [c_{33}(\omega) - c_{44}(\omega)]^2$ (see Appendix B). Therefore, for zero anisotropy $c_{23}(\omega) + c_{44}(\omega)$ must be equal to $c_{33}(\omega) - c_{44}(\omega)$. The function $c_{44}(\omega)$ is constant across the whole frequency range, $c_{23}(\omega)$ is strictly decreasing with frequency and $c_{33}(\omega)$ is strictly increasing with frequency (Figure 4). At certain frequency (here it is at
$\approx 10^4$ Hz), the $c_{33}$ and $c_{23}$ components are in such combination that $c_{23}(10^4) + c_{44}(10^4) \approx c_{33}(10^4) - c_{44}(10^4)$, so $\delta^{YZ} = 0$ and the P-wave velocity in the $YZ$ plane behaves as in a fully isotropic media.

In the $XZ$ plane, $\epsilon^{XZ}$ and $\delta^{XZ}$ are frequency dependent in the model with connected cracks. $\epsilon^{XZ}$ exhibits moderate elastic anisotropy at low frequencies while $\delta^{XZ}$ exhibits moderate elastic anisotropy at high frequencies. Other parameters are frequency independent and exhibit certain non-zero value from weak to moderate elastic anisotropy.




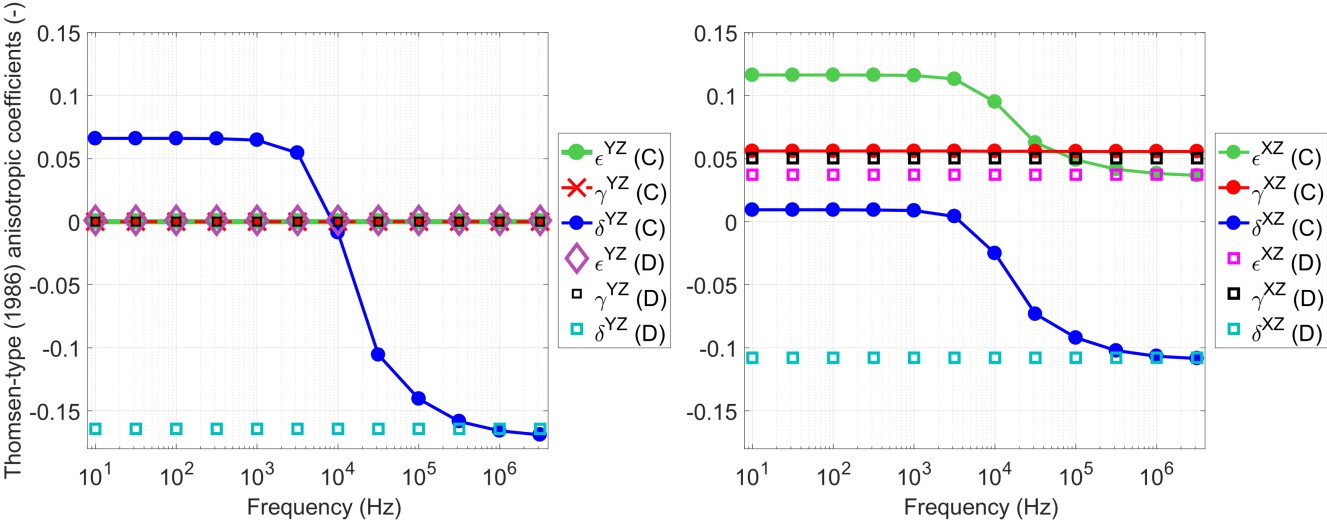

**Figure 8.** Thomsen-type anisotropic parameters in the $YZ$ (left) and $XZ$ (right) planes.

### 4.4.2 The universal elastic anisotropy index

The universal elastic anisotropy index $A^U$ (Ranganathan and Ostoja-Starzewski, 2008) is widely used to measure the anisotropy strength in crystallography, engineering and material science. This parameter is designed to evaluate the anisotropy strength of crystals having any elastic symmetry class (Ranganathan and Ostoja-Starzewski, 2008). Since $A^U$ is a scalar, it gives a simple and fast identification of the overall anisotropy strength of a model. $A^U = 0$ corresponds to zero anisotropy of a model while the discrepancy of $A^U$ from zero defines the anisotropy strength and accounts for both the shear and the bulk contributions simultaneously. In analogy to the universal elastic anisotropy index, two other parameters are adopted which define the anisotropy strength in bulk $A^{bulk}$ and in shear $A^{shear}$. As far as we are concerned, these parametrs have not been widely used in earth sciences, only a few studies were found. Almqvist and Mainprice (2017) applied the universal elastic anisotropy index and similar two parameters for bulk and shear to study seismic properties and anisotropy of the continental crust. Kube and De Jong (2016); Duffy (2018); Vieira et al. (2019) applied $A^U$ to quantify the elastic anisotropy of polycrystals. A brief review of these anisotropic measures and all necessary equations for their calculation are provided in Appendix C.

Figure 9 shows the universal elastic anisotropy index $A^U$ and the anisotropy measures in bulk $A^{bulk}(\omega)$ and shear $A^{shear}$. For the model with disconnected cracks, $A^U$ is constant and frequency independent (Figure 9, black line). Because $A^U$ has a certain small value (about 0.058), the model with disconnected cracks exhibit a certain small anisotropy. For the model with connected cracks, $A^U$ in the high frequency limit is almost the same as for the model with disconnected cracks (Figure 9, red line) (the nature of the discrepancy is related to the region containing the crack intersection and explained in the previous section). For the model with connected cracks, the overall anisotropy slightly decreases towards lower frequencies until $10^{4.3}$ Hz, reaching its minimum of 0.048 (Figure 9, red line). This local minimum indeed corresponds to the $c_{23}$ attenuation peak





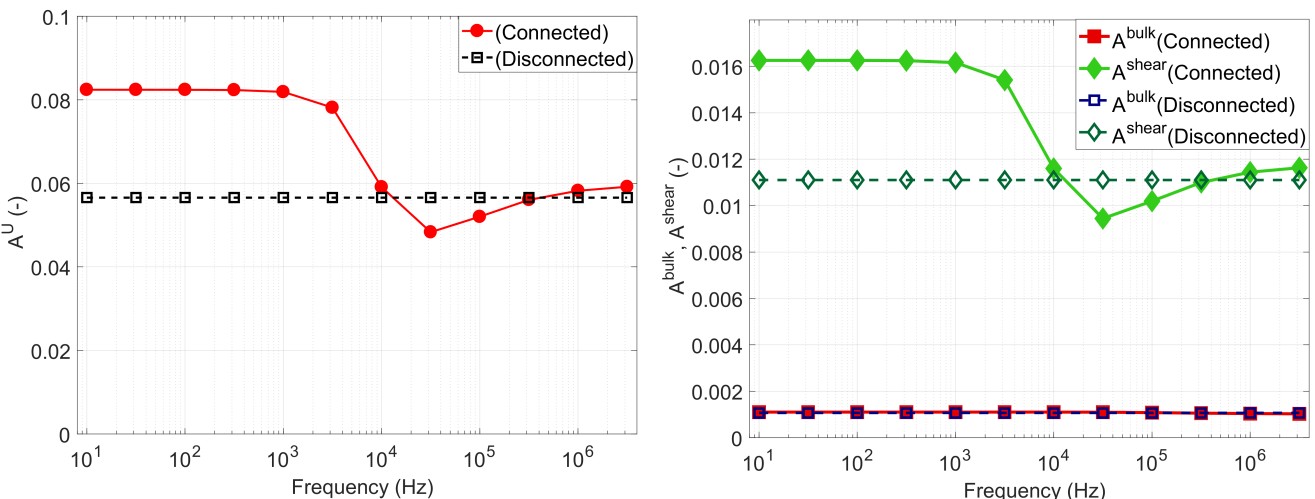

**Figure 9.** The universal elastic anisotropy index measure $A^U$ versus frequency (left) and the anisotropy measures in bulk and shear ($A^{bulk}$ and $A^{shear}$) (right). These plots show that the overall anisotropy if the model increases in the low frequencies due to fluid flow.

(Figure 4, (c)). Then, still towards lower frequencies, $A^U(\omega)$ increases reaching its maximum of 0.083 at frequencies below

$10^3$ Hz (Figure 9, red line). Thus, the overall anisotropy of the model mainly increases due to squirt flow between the cracks, so the cracks connectivity increases the overall anisotropy of the model towards low frequencies.

The anisotropy measure in bulk $A^{bulk}$ is constant and frequency independent for the models with connected and disconnected cracks (Figure 9 , right). It means that fluid flow do not affect bulk properties of the model neither the anisotropy strength in bulk. On the other hand, the anisotropy measure in shear $A^{shear}(\omega)$ basically reproduces the behavior of the universal elastic

anisotropy index measure $A^U$. Therefore, one can conclude that the fluid flow changes anisotropy in shear mode, but not in bulk mode.

## 5  Discussion

### 5.1  Elastic anisotropy

Thomsen-type anisotropic parameters provide a very detailed description of the velocity anisotropy in different planes. Most

importantly, only a limited number of the stiffness tensor coefficients is needed to calculate $\epsilon, \delta, \gamma$ in each plane. Thus, Thomsen parameters can be used to quantify the medium anisotropy using seismic data. On the other hand, when all components of the stiffness tensor are known and the model's symmetry is low, it is difficult to analyse the overall anisotropy due to a large number of Thomsen parameters. For example, if the model exhibits orthorhombic symmetry, one should analyze nine Thomsen anisotropic parameters (three in each plane). Due to a large number of Thomsen parameters in this study, it is difficult

to evaluate whether the overall medium's anisotropy is increasing or decreasing with frequency and how far the current model





is from the closest isotropy. Thus, in addition to (or instead of) the Thomsen-type anisotropic parameters, the universal elastic anisotropy index can be used. The universal elastic anisotropy index $A^U$ and the related measures in bulk $A^{bulk}$ and shear $A^{shear}$ provide the overall description of the anisotropy strength regardless of the model's complexity. The calculation of these parameters is as simple as the calculation of the Thomsen parameters. An obvious disadvantage of the universal elastic

anisotropy index (and related measures) is that it requires knowledge of the full stiffness tensor. Thus, this anisotropic measure can be useful to evaluate results of numerical simulations, of laboratory experiments and for measuring the anisotropy of single crystals.

The analysis of two sets of anisotropic measures shows that (i) the the overall anisotropy of the model with connected cracks (Figure 1) mainly increases due to squirt flow towards low frequencies with a slight local decrease at intermediate frequencies

(Figure 9 (left)), (ii) in the $YZ$ plane, the magnitude of the "delta" anisotropy parameter decreases, reaches zero and then increases again (reaching $\approx 1/3$ of its high frequency value) towards low frequencies (Figure 8 (left), blue curve) and (iii) in the $XZ$ plane, the "delta" anisotropy parameter decreases towards low frequencies (Figure 8 (right), blue curve) while the "epsilon" anisotropy parameter increases (Figure 8 (right), green curve).

### 5.2 Comparison against previous works

In this study, we numerically solve a coupled fluid-solid deformation problem at the pore scale. If we consider the mesoscopic scale scenario and use Biot's (1941) equations, the fluid flow effects on the effective moduli are equivavalent to that of the coupled elastic-Stokes equations (as in the present study), as it was shown by Quintal et al. (2016). The frequency-dependent anisotropy due to fluid flow at the mesoscopic scale for orthogonal fracture sets with different degrees of connectivity was numerically studied by Rubino et al. (2017) but their study was limitted to two dimensions. The main conclusion of Rubino

et al. (2017) is that the anisotropy decreases with fracture connectivity in the seismic frequency band due to fluid flow between connected fractures. The $YZ$-plane in the present 3D numerical model is resonably equivalent to the 2D numerical model of Rubino et al. (2017) as well as the physical mechanism under consideration. The results of Rubino et al. (2017) are reflected in Figure 5 of this study. However, a more in depth analysis shows that the anisotropy in the $YZ$-plane decreases, reaches zero and then increases again towards low frequencies due to squirt flow (Figure 5 (left), green, yellow and blue curves, respectively).

Moreover, our present study shows that the overall anisotropy of the model with cracks of finite length actually increases due to fluid flow between interconnected cracks (Figures 8, 9). This conclusion is not universal and is valid only for a specific set of model parameters.

Barbosa et al. (2017) performed a more detailed study of seismic anisotropy for a similar fracture geometry in two dimensions, as in the study of Rubino et al. (2017), specifying that the decrease in anosotropy is described by the anisotropy parameter

$\delta$ while $\epsilon$ is zero. Furthermore, they observed a polarity change of the P-wave phase velocity behavior with frequency. In the present study, the "delta" anisotropy parameter in the $YZ$-plane is more pronounced in the low frequency limit (Figure 5 (left), blue curve) compared to the work of Barbosa et al. (2017) due to different material properties and the three-dimensional nature of the present model configuration.





In summary, fluid flow effects on seismic anisotropy are non-linear with a possible increase and decrease in the elastic
anisotropy at different frequencies. These two extreme cases, the maximum negative and the maximum positive $\delta$ parameter
(and, hence, P-wave velocity) in the $YZ$ plane, correspond to the relaxed and unrelaxed states. In the relaxed state, one can cal-
culate the effective dry elastic moduli and use Gassmann's equations to obtain the effective moduli of the saturated medium. In
the unrelaxed state, one can calculate the effective elastic moduli by restricting fluid flow (by using zero displacement bound-
ary conditions in the cracks intersections). In other words, seismic anisotropy may behave completely different in different
scenarios, therefore, more studies should be performed, especially with the sensitivity analysis of model parameters.

## 6 Conclusions

We have numerically calculated the frequency-dependent elastic moduli of a fluid-saturated porous medium caused by squirt
flow. The considered 3D numerical models consist of two perpendicular connected or disconnected cracks embedded in a solid
grain material. Cracks are represented by very flat cylinders filled with a fluid. Grains are described as a linear isotropic elastic
material while the fluid phase is described by the quasistatic linearized compressible Navier-Stokes momentum equation.

We showed that seismic velocities are azimuth-, angle- and frequency-dependent due to squirt flow between connected
cracks. The resulting elastic frequency-dependent anisotropy was analylized by using the Thomsen-type anisotropic parameters
and the universal elastic anisotropy index. The latter is a scalar parameter which can be used to analyse the overall anisotropy
of the model and its divergence from the closest isotropy. We showed that the seismic anisotropy may locally decrease as well
as increase due to squirt flow in one specific plane. However, the overall anisotropy of the model mainly increases due to squirt
flow between the cracks towards low frequencies. In the model with disconnected cracks, no fluid flow occurs and, thus, the
effective properties of the model correspond to the elastic limit. Seismic velocities are only azimuth- and angle- dependent as
for a fully elastic material and they are independent of frequency. The elastic limit is equivalent to the high frequency limit for
the model with connected cracks.
Our conclusion is that squirt flow do affect effective elastic properties of cracked rocks and, thus, seismic velocity anisotropy.
Given that seismic anisotropy variations with frequency are very sensitive to the pore space geometry and material properties,
we cannot make a very general prediction.

## Appendix A: Boundary conditions for $c_{ij}$ off-diagonal components

Let's consider a cuboid, volume $V = (0, Lx) \times (0, Ly) \times (0, Lz)$ and $\Gamma$ its boundary $\Gamma = \Gamma^{xz0} \cup \Gamma^{xzL} \cup \Gamma^{yz0} \cup \Gamma^{yzL} \cup \Gamma^{xy0} \cup \Gamma^{xyL}$,
where, for example, $\Gamma^{xz0}$ represents a $xz$ plane with zero coordinate and $\Gamma^{xzL}$ represents a $xz$ plane with $Ly$ coordinate etc.
There are six planes in total.

The mixed test for the $c_{13}$ component can be derived from the anisotropic stress-strain relation (Hooke's law) (similarly to
the $c_{13}$ component in a VTI medium (Alkhimenkov et al., 2019)).





$\Gamma^{xyL}$ is set to $u_{zz} = \Delta u$; $u_{xx}, u_{yy}$ are free

   $\Gamma^{yzL}$ is set to $u_{xx} = \Delta u$; $u_{zz}, u_{yy}$ are free,

where $\Delta u = 10^{-6}$. In other four planes, the normal component of the displacement is set to zero, other components are free. The the stress-strain relation for the $\langle \sigma_1 \rangle$ stress component is

$$\langle \sigma_1 \rangle = c_{11} \langle \epsilon_1 \rangle + c_{12} \langle \epsilon_2 \rangle + c_{13} \langle \epsilon_3 \rangle. \tag{A1}$$

Using the described above boundary conditions and setting $\langle \epsilon_2 \rangle = 0$, equation (A1) becomes

$$\langle \sigma_1 \rangle = c_{11} \langle \epsilon_1 \rangle + c_{13} \langle \epsilon_3 \rangle. \tag{A2}$$

Equation (A2) can be solved for the $c_{13}$ component, the solution is

$$c_{13} = \frac{\langle \sigma_1 \rangle}{\langle \epsilon_1 \rangle} - c_{11}. \tag{A3}$$

Equation (A3) is used to calculate the $c_{13}$ component ($c_{11}$ is taken from the direct tests).

The mixed test for the $c_{23}$ (in this numerical model, the $c_{23}$ is dispersive) component again can be derived from the anisotropic stress-strain relation (Hooke's law) (similarly to the previous test).

   $\Gamma^{xyL}$ is set to $u_{zz} = \Delta u$; $u_{xx}, u_{yy}$ are free

$\Gamma^{xzL}$ is set to $u_{yy} = \Delta u$; $u_{zz}, u_{xx}$ are free

In other four planes, the normal component of the displacement is set to zero, other components are free. Then, using the following equation

$$c_{23} = \frac{\langle \sigma_2 \rangle}{\langle \epsilon_2 \rangle} - c_{22}, \tag{A4}$$


the $c_{23}$ component is calculated ($c_{22}$ is taken from the direct test). Equations (A3)-(A4) are found from the Hooke's Law considering non-zero strains in $x$- and $z$ (in $y$- and $z$) directions and, then, solving a system of two equations analytically.

**Appendix B: Thomsen-type anisotropic parameters**

Thomsen-type anisotropic parameters (Thomsen, 1986) are widely used in the applied geophysics community. Thomsen weak
anisotropy parameters were originaly developed for vertical transverse isotropic media (Thomsen, 1986). A natural extension of these parameters to orthorhombic media was suggested by Tsvankin (1997); Bakulin et al. (2000a). These parameters





correspond to the anisotropy of the compression and shear waves in orthorhombic media in different Cartesian propagation planes. In the $YZ$-plane, Thomsen-type anisotropic parameters are

$$\epsilon^{(YZ)}(\omega) = \frac{c_{22}(\omega) - c_{33}(\omega)}{2c_{33}(\omega)}, \quad \gamma^{(YZ)}(\omega) = \frac{c_{66}(\omega) - c_{55}(\omega)}{2c_{55}(\omega)}, \tag{B1}$$


and

$$\delta^{(YZ)}(\omega) = \frac{[c_{23}(\omega) + c_{44}(\omega)]^2 - [c_{33}(\omega) - c_{44}(\omega)]^2}{2c_{33}(\omega)[c_{33}(\omega) - c_{44}(\omega)]} \tag{B2}$$

In the $XZ$-plane, Thomsen-type anisotropic parameters are

$\quad \epsilon^{(XZ)}(\omega) = \dfrac{c_{11}(\omega) - c_{33}(\omega)}{2c_{33}(\omega)}, \quad \gamma^{(XZ)}(\omega) = \dfrac{c_{66}(\omega) - c_{44}(\omega)}{2c_{44}(\omega)},$ $\qquad\qquad$ (B3)

and

$$\delta^{(XZ)}(\omega) = \frac{[c_{13}(\omega) + c_{55}(\omega)]^2 - [c_{33}(\omega) - c_{55}(\omega)]^2}{2c_{33}(\omega)[c_{33}(\omega) - c_{55}(\omega)]}, \tag{B4}$$

**Appendix C: The universal elastic anisotropy index parameter**

Assuming that one deals with an anisotropic frequency-dependent effective 4-th rank stiffness tensor $\mathbf{C}$ (might be frequency dependent $\mathbf{C} \Rightarrow \mathbf{C}(\omega)$), a compliance tensor is defined as $\mathbf{S}(\omega) = \mathbf{C}(\omega)^{-1}$. Then, for each frequency, the effective single orientation 4-th rank stiffness and compliance tensors are uniformly distributed and the isotropic stiffness and compliance tensors are calculated. Averaging the single orientation stiffness tensor belongs to the Voigt assumption which is the theoretical maximum value of the isotropic elastic moduli. On the other hand, averaging the single orientation compliance tensor belongs to

the Reuss assumption which is the theoretical minimum value of the isotropic elastic moduli. The resulting isotropic tensors can be expressed in terms of the spherical and deviatoric parts corresponding to bulk $K$ and shear moduli $\mu$:

$$\mathbf{C}^V(\omega) = 3K^V \mathbf{J} + 2\mu^V \mathbf{D} \tag{C1}$$

and

$\quad \mathbf{S}^R(\omega) = \dfrac{1}{3K^R}\mathbf{J} + \dfrac{1}{2\mu^R}\mathbf{D},$ $\qquad\qquad$ (C2)





where the superscripts $"^{V}"$ and $"^{R}"$ correspond to Voigt and Reuss estimates, respectively. $\mathbf{J}$ and $\mathbf{D}$ are the spherical (volumetric) and deviatoric parts of the symmetric unit 4-th order tensor.

The double contraction the scalar product (quadruple contraction) of Equation (C1) and Equation (C2) gives

$$\mathbf{C}^V(\omega) :: \mathbf{S}^R(\omega) = \frac{K^V(\omega)}{K^R(\omega)} + 5\frac{\mu^V(\omega)}{\mu^R(\omega)}, \tag{C3}$$


If the effective stiffness tensor is isotropic, then $\mathbf{C}^V(\omega) = \left(\mathbf{S}^R(\omega)\right)^{-1}$ and $K^V/K^R = \mu^V/\mu^R = 1$. Therefore, when the effective stiffness tensor is isotropic, the value of Equation (C3) equals to 6 and this value increases when the effective stiffness tensor becomes anisotropic. Thus, the universal elastic anisotropy index measure $A^U$ is defined as (Ranganathan and Ostoja-Starzewski, 2008):

$$A^U(\omega) = \frac{K^V(\omega)}{K^R(\omega)} + 5\frac{\mu^V(\omega)}{\mu^R(\omega)} - 6 \geq 0, \tag{C4}$$

In geophysics, the separation of the elastic anisotropy measure in bulk and shear modes is necessary because rocks might exhibit different frequency-dependence due to bulk and shear deformations. Therefore, in analogy to the universal elastic anisotropy index measure $A^U$, the anisotropy measures in bulk $A^{bulk}(\omega)$ and shear $A^{shear}(\omega)$ can be defined

$$A^{bulk}(\omega) = \frac{K^V(\omega)}{K^R(\omega)} - 1 \tag{C5}$$

and

$$A^{shear}(\omega) = \frac{G^V(\omega)}{G^R(\omega)} - 1 \tag{C6}$$

These two parameters $A^{bulk}(\omega)$ and $A^{shear}(\omega)$ obey the same interpretation as the universal elastic anisotropy index measure: $A^{bulk} = 0$ ($A^{shear} = 0$) corresponds to zero bulk (shear) anisotropy of the model while the discrepancy of $A^U$ from zero anisotropy defines the anisotropy strength in bulk (shear).

The Voigt and Reuss estimates ($K^V$, $K^R$, $\mu^V$ and $\mu^R$) can be calculated via simple algebraic formulas for different symmetry classes which can be found elsewhere, e.g in Ravindran et al. (1998) for orthorhombic symmetry, in Feng et al. (2012) for
tetragonal symmetry and in Duffy (2018) for cubic symmetry. Thus, for orthorhombic symmetry (lowest possible symmetry created by two perpendicular sets of cracks embedded into an isotropic material), the Voigt and Reuss bulk moduli for are (written for the components of stiffness (compliance) matrices $c_{ij}(s_{ij})$, in Voigt notation)

$$K^V = \frac{1}{9}\left[(c_{11} + c_{22} + c_{33}) + 2(c_{12} + c_{13} + c_{23})\right] \tag{C7}$$





and

$$K^R = \left[ (s_{11} + s_{22} + s_{33}) + 2\,(s_{12} + s_{13} + s_{23}) \right]^{-1}. \tag{C8}$$

Similarly, the Voigt and Reuss shear moduli are (in Voigt notation)

$$\mu^V = \frac{1}{15} \left[ (c_{11} + c_{22} + c_{33} - c_{12} - c_{13} - c_{23}) + 3\,(c_{44} + c_{55} + c_{66}) \right] \tag{C9}$$

and

$$\mu^R = 15 \left[ 4\,(s_{11} + s_{22} + s_{33} - s_{12} - s_{13} - s_{23}) + 3\,(s_{44} + s_{55} + s_{66}) \right]^{-1}. \tag{C10}$$

Equations (C7)-(C10) are valid for orthorhombic symmetry and for higher symmetries: tetragonal, transverse isotropy and

cubic. Thus, for evaluating the universal elastic anisotropy index $A^U$ and the anisotropy measures in bulk $A^{bulk}(\omega)$ and shear $A^{shear}(\omega)$, one can use equations (C7)-(C10) to calculate the Voigt and Reuss estimates ($K^V$, $K^R$, $\mu^V$ and $\mu^R$) and, then, calculate $A^U$ using equation (C4) and $A^{bulk}(\omega)$ and $A^{shear}(\omega)$ using equations (C5) and (C6), respectively.

*Author contributions.* YA performed the numerical simulations and wrote the manuscript. The idea of this project was first inspired by the paper of Rubino et al. (2017). A detailed project was planed by YA and BQ. EC, SL and BQ provided many ideas and suggestions which

influenced the project path and helped writing the manuscript.

*Competing interests.* The authors declare that they have no conflict of interest.

*Acknowledgements.* This research is funded by the Swiss National Science Foundation, project number 172691. Yury Alkhimenkov thanks J. Germán Rubino (CONICET, Centro Atómico Bariloche, Argentina) for fruitful discussions on the frequency-dependent anisotropy due to fluid flow. We thank Nicolás D. Barbosa (University of Geneva, Switzerland) for fruitful discussions regarding the polarity change of the P-

wave velocity with frequency. We thank Irina Bayuk (Russian Academy of Sciences, Russia) for useful discussions regarding the fourth-rank tensors averaging and the elastic symmetry classes.



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
