# Peer review of "Azimuth-, angle- and frequency-dependent seismic velocities of cracked rocks due to squirt flow"

_Solid Earth, 2019_

## Referee Comment (RC1) · Vishal Das (Referee) · 23 Jan 2020

**Review comments on *Azimuth-, angle-, frequency-dependent seismic velocities of cracked rocks due to squirt flow**

In this work, the authors use numerical methods to study the azimuth, angle and frequency dependent seismic velocities for cracked rocks due to squirt flow. The authors do a rigorous 3D numerical study using a simple geometry and set up two different experiments – connected and disconnected cracks. The authors show that the seismic velocities are dependent on azimuth, angle and frequency only for connected cracks case. The authors also compare the results of the disconnected crack with the high frequency results of the connected cracks case. Apart from Thomsen's parameter that is primarily used for anisotropic studies in geophysics, the authors also formulate and use scalar parameter called anisotropy index parameter to describe the anisotropy dependence for squirt flow.

This is a nice piece of numerical study for a simple geometry that gives new insights into anisotropy that arises due to squirt flow. The study is missing benchmarking of the numerical results with some simple analytical solutions or to some laboratory data. Benchmarking will help the readers have more confidence on the numerical results. I believe that the authors could make some comments on the practical uses of this in some details apart from the introduction section of the paper where they mention a generic statement on it. Also, I think that the value of the work can be increased by application of the work on a real digital rock sample apart from the geometry that has been considered. The authors can also think about presenting the figures for the results section in a better/simpler way (if possible). These figures are the crux of the paper and will be the most important thing for the readers to gain as a take-home message.

I have mentioned my specific comments on the paper in the following section.

**General comments:**

Was any software (eg. COMSOL) used for the simulations?

Can there be a way to apply this work for a real 3D digital rock sample?

Can the results be benchmarked with some standard analytical solution or measurements from the laboratory?

**Specific comments:**

**Comments on Introduction section:**

1st line is a good motivation for the study but leaves the reader with the question why or how cracked rocks play a crucial role. Some references or a line to elaborate it would be effective. One might argue that the next one mentions hydraulic properties are affected by cracks is the justification, however, I still think it deserves some clarity.

Line 24 – Mesoscopic scale definition preferred before this line.

Missing reference to Mavko and Nur (1975) classic paper that introduced "squirt flow"!

Line 51 – Das et al. (2019) Numerical simulation of coupled fluid-solid interaction at the pore scale: A digital rock-physics technology is the full paper in Geophysics and is a better reference than the conference abstract.

**Comments on Numerical methodology section:**

Line 79 – Definition of $u$?

In the finite element numerical solver are you solving for displacement or velocity for the fluid phase (equation 3)? How is the coupling done in terms of displacement at the boundary between the two phases? It might be useful to mention a few lines here instead of just references.

Line 86 – Why are the inertial terms neglected? What would be the limitations that might arise due to this approximation? Any discussion on how this can be included?

Line 88 – Why is the PARDISO solver used? The reader might wander what are the advantages of using this versus any other solver.

Line 89 – What is direct relaxation test? Reference/ one line description can be useful.

Line 96 – Is the harmonic displacement a function of time? Are you solving the equations in time domain or frequency domain? How does combining equations 2 and 3 work if the equations are solved in time domain vs frequency domain?

Line 100 – What are mixed direct tests?

**Comments on Numerical model:**

Line 108 Any specific reason for having an aspect ratio of 0.01?

Line 109 Any specific reason for using glycerol?

Line 110 In Table 1, how is the Bulk Modulus (K) of the fluids used in equation 3?

Line 111 Can there be some quantitative ways of describing fine and coarser mesh – size of the elements (maximum and minimum size, average size of the mesh element) including the details mentioned in the caption of Figure 2. Also, along the z axis of the mesh, parts of it has 3 elements as per the snapshot in Figure 2. Is this sufficient number of elements from numerical point of view? Is there any way to justify that the numerical solution is stable with the number of elements used?

Line 113 What type of system configuration was used for making the simulations? It seems like 0.95TB of RAM would require very specific type of machine to run. How much time did it take to run each of the simulation?

Line 126-127 Does the difference between the stiffness of the two disconnected cracks also depend on the boundary conditions applied to the system? In other words, will the stiffness be always different irrespective of the boundary condition applied?

**Comments on Results section:**

Line 147 Unclear what is meant by the cracks can be described by only two compliances as per equation 6?

I am also confused that in Line 142 it is mentioned that there is a significant difference due to vertical crack separation. However, in Line 148, it is mentioned that only connected cracks case is considered? Can this be explained better?

Line 165 – What are the values of frequencies used for LF, Fc and HF?

Line 173 – It is unclear what is meant by the elastic limit in this case? Can this be compared to any standard elastic limits that are mentioned in literature?

Figure 4 might be improved from a reader's perspective.

Line 180 – Even in figure 4a, the Real part of the c22 and c33 component coincides. Is there a specific reason for pointing out separately the attenuation and dispersion components are same due to symmetry of the model?

Line 183 – It might be useful to explain what negative attenuation means in physical sense. Is there a reason for the negative attenuation behavior?

Line 194 – It might be useful to mention how the seismic velocities are calculated as a function of phase?

Figure 6 – Is there a reason for the discrepancy between V_SV between the disconnected crack model and the high frequency result at a phase angle of 0, 90 and 180 degrees?

**Comments related to Discussions:**

Line 311 – What are the model parameters that the conclusion will depend upon? The authors mention about the need of sensitivity analysis of the model parameters. It would be useful to give a qualitative idea about the possible model parameters that the results will depend on.

---

## Referee Comment (RC2) · Yves Gueguen (Referee) · 23 Jan 2020

Solid Earth se-2019-176

Azimuth-, angle- and frequency-dependent seismic velocities of cracked rocks due to squirt flow

by

Yury Alkhimenkov, Eva Caspari, Simon Lissa, and Beatriz Quintal
* * *
The paper is focused on a numerical model aiming at calculating the frequency and

anisotropic response of a saturated cracked rock for a passing seismic wave. The model provides a numerical calculation following the previous work of B. Quintal. The model considers two orthogonal thin cracks embedded in a homogenous background. Two situations are examined: the two cracks are either connected or disconnected. The numerical method has been previously used by Quintal (2016, 2019). It consists in applying relaxation tests to a viscoelastic medium. The CIJ constants are obtained by averaging.

Several remarks are the following. First the frequency dependent curves extend over a broad frequency range, for a unique crack aspect ratio. This implies that squirt flow would never be focused on a narrow frequency range (unless the system size plays a dominant role in the calculation). This remark is important for the geophysical implications: Fig. 4a shows that the width of the attenuation peak (at half amplitude) is about one order and half magnitude. Second, the model considers two cracks of 0.1 m in a cube of 0.24 m size. In terms of crack density, this means a very high crack density (close to 1). This is consistent with the large decrease of C22 and C33 in the dry case, compared to the original values of the intact rock (table 1). But such high values are not realistic. Third, Fig. 4c shows a negative 1/Q and a very high dispersion for C23. Probably, the negative sign (which is unphysical) is an error of convention. But the high value (almost 0.4 for 1/Q) should be related to the very high crack density (i.e. the size of the system) and the low value of C23. Four, although a precise comparison is impossible, it would be of interest to discuss these results against (effective medium) calculations published some years ago (Guéguen, Y. , and Sarout, J., 2009. Crack-induced anisotropy in crustal rocks : predicted dry and fluid-saturated Thomsen's parameters. Physics of the Earth and Planetary Interiors, 172, 116-124; and Guéguen, Y. , and Sarout, J., 2011. Characteristics of anisotropy and dispersion in cracked medium. Tectonophysics, 503, 1-2, 165-172.) In both cases, the goal is similar but the methods differ. In terms of anisotropic compliances dispersion, it seems (from a first check) that the predictions of GS agree with the present results. They give a prediction of dispersion for Sijkl in terms of the two crack density tensors alpha and

beta. Non-zero values are predicted only if the i,j,k,l index is 2 or 3 (given the cracks orientations in the present case).

In conclusion, this is an interesting paper.

25 January 2020 Yves Guéguen

---

## Author Comment (AC1) · 20 Mar 2020

**Referee's comment 1**

The paper is focused on a numerical model aiming at calculating the frequency and anisotropic response of a saturated cracked rock for a passing seismic wave. The model provides a numerical calculation following the previous work of B. Quintal. The model considers two orthogonal thin cracks embedded in a homogenous background. Two situations are examined: the two cracks are either connected or disconnected. The numerical method has been previously used by Quintal (2016, 2019). It consists in applying relaxation tests to a viscoelastic medium. The CIJ constants are obtained
by averaging.

**Author's reply 1**

Dear Reviewer,

Thank you very much for the time you have dedicated to review and comment our manuscript. We believe that your comments have helped us to improve significantly the quality of the work. Please find blow the responses to your comments.

Kind regards, Yury Alkhimenkov, on behalf of the authors

**Changes in the manuscript 1**

-

**Referee's comment 2**

Several remarks are the following. First the frequency dependent curves extend over a broad frequency range, for a unique crack aspect ratio. This implies that squirt flow would never be focused on a narrow frequency range (unless the system size plays a dominant role in the calculation). This remark is important for the geophysical implications: Fig. 4a shows that the width of the attenuation peak (at half amplitude) is about one order and half magnitude.

**Author's reply 2**

That's indeed true. Although several analytical solutions predict a narrow frequency range, for example, Gurevich et al., (2010) and Collet and Gurevich (2016), we published results showing a broad frequency range for squirt flow in a different paper [Alkhimenkov et. al., 2020] considering a different pore-space geometry. For the pore-space geometry presented in the current manuscript we again consistently observe that squirt flow is not focused on a narrow frequency range.

**Changes in the manuscript 2**
page 10, line 207-209:

"Note that the width of the inverse quality factor peak (at half amplitude) for the components c22 and c33 is about one order and half magnitude (Figures 4a and 4c). It means that attenuation and dispersion due to squirt flow play a significant role over a broad frequency range even for cracks with a single aspect ratio."

**Referee's comment 3**

Second, the model considers two cracks of 0.1m in a cube of 0.24 m size. In terms of crack density, this means a very high crack density (close to 1). This is consistent with the large decrease of C22 and C33 in the dry case, compared to the original values of the intact rock (table 1). But such high values are not realistic.

**Author's reply 3**

Yes, indeed, the crack density is quite high. This is a synthetic study and all physical effects will be the same for a medium with a smaller crack density (the magnitude will be smaller). According to our calculations, our crack density is closer to 0.1. p = 1/ 0.24^3 *0.1^3 *2 = 0.1447, where 0.24m – a cuboid size, 0.1m - the radius of the crack, 2 - the number of the cracks. This is a correct formula for cracks with random orientations [Bristow, J. R., 1960; Kachanov, M., & Mishakin, V. V. 2019]. For our model geometry, this definition can be considered as a rough approximation. A better definition for our geometry requires the usage of the crack density tensor (second-order) plus another fourth order tensor [Kachanov, M., & Mishakin, V. V. 2019].

**Changes in the manuscript 3**

-

**Referee's comment 4**

Third, Fig. 4c shows a negative 1/Q and a very high dispersion for C23. Probably, the negative sign (which is unphysical) is an error of convention. But the high value (almost

0.4 for 1/Q) should be related to the very high crack density (i.e. the size of the system) and the low value of C23.

**Author's reply 4**

Yes, indeed the dispersion as well as "attenuation" for the C23 component is strong. First, the negative sign for "attenuation" is correct. We are not the first who observe this negative ratio of Im(C23)/Re(C23). For example, Guo et. al. (2017) already reported a negative sign in their results of 2D numerical simulations. Let us first discuss the negative sign and, then, why the value of 1/Q is high.

The C23 component is the coupling component between C22 and C33 components. In the stiffness tensor, we do have high and low limits for components which are based on the energy constraints. When we are talking about wave dispersion and attenuation in anisotropic media, we should consider all stiffness components which are needed to calculate wave speeds and attenuations. The C23 component does not enter any wave mode in any direction alone, the C23 component is always used together with C22 or/and C33, which have high positive values of attenuation. Therefore, no wave will gain energy. In other words, different components of the stiffness tensor might have positive or negative values of "attenuation" but when we calculate the velocity and attenuation of a wave, the cumulative effect of all Cij components must be physical (it can be seen in Figures 5-7) and no negative attenuation will be observed. This negative "attenuation" sign for C23 was also verified using Kramers-Kronig relations. We think that there is a terminology issue. In fact, we have a negative value of the ratio Im(C23)/Re(C23), but it might be misleading to call that attenuation, which is associated with the characteristic of a wave following a definition based on energy-related considerations.

The high negative value of Im(C23)/Re(C23) is due to the low values of C23 in figure 4c (17 GPa or 7 GPa). It is related to the high crack density in our model geometry but for a model with lower crack density, we think, that the negative value of Im(C23)/Re(C23)

will be also high.

Now in the manuscript, we refrain from using the terminology "attenuation" for the ratio Im(C23)/Re(C23). We now simply refer to it as a ratio and we reserve the terminology attenuation for wave modes. Furthermore, the wave modes as explained above won't ever show a negative attenuation.

**Changes in the manuscript 4**

Page 4, line 118-120: "Note that usually the inverse quality factor is used as a measure of attenuation (O'connell and Budiansky, 1978). In this study, we show the inverse quality factor for each component of the stiffness tensor, even though the ratio Im(cij(omega))=Re(cij(omega)) does not represent attenuation of any corresponding wave mode for some components."

Page 9, lines 199-206: "The c12 and c13 components are non-dispersive, the c23 component exhibits strong negative dispersion and a negative inverse quality factor peak shifted towards high frequencies compared to that of the c22, c33 components. A similar phenomenon has been reported by Guo et al. (2017) in the context of two-dimensional simulations. The c23 component does not correspond to a wave mode alone, it is always used together with c22 or/and c33 components. Therefore, no wave will gain energy. This negative inverse quality factor sign for the c23 component was also verified using Kramers-Kronig relations. In other words, different components of the stiffness tensor might have positive or negative values of the ratio Im(c23)=Re(c23) but when we calculate the velocity and the inverse quality factor of a wave, the cumulative effect of all cij components must be physical and no negative attenuation will be observed."

**Referee's comment 5**

Four, although a precise comparison is impossible, it would be of interest to discuss these results against (effective medium) calculations published some years ago

(Guéguen, Y. , and Sarout,J., 2009. Crack-induced anisotropy in crustal rocks: predicted dry and fluid-saturated Thomsen's parameters. Physics of the Earth and Planetary Interiors, 172, 116-124; and Guéguen, Y. , and Sarout, J., 2011. Characteristics of anisotropy and dispersion in cracked medium. Tectonophysics, 503, 1-2, 165-172.) In both cases, the goal is similar but the methods differ. In terms of anisotropic compliances dispersion, it seems (from a first check) that the predictions of GS agree with the present results. They give a prediction of dispersion for Sijkl in terms of the two crack density tensors alpha and beta. Non-zero values are predicted only if the i,j,k,l index is 2 or 3 (given the cracks orientations in the present case).

**Author's reply 5**

We compared the results of our numerical solver against an analytical solution for squirt flow in a recently published manuscript [Alkhimenkov et. al., 2020]. There is no analytical solution for the presented model geometry, only the low and high-frequency limits can be described analytically. Guéguen, and Sarout, (2009, 2011) presented analytical models considering poroelastic and squirt flow effects at low and high-frequency limits. They observe that the anisotropy (described by Thompson's parameters) is, in general, more pronounced at high frequencies than at low frequencies.

**Changes in the manuscript 5**

page 2, line 47-50: "There are several analytical solutions for squirt flow (O'Connell and Budiansky, 1977; Dvorkin et al., 1995; Chapman et al., 2002; Guéguen and Sarout, 2009, 2011; Gurevich et al., 2010) which are based on simplified pore geometries and many physical assumptions."

Page 17, line 352-365 (new subsection):

"A qualitative comparison against analytical models

Numerical simulations are useful but analytical models are especially attractive since they help us to better understand the physics and do not require sophisticated numerical simulations. The limitations of the analytical solutions are restricted to simple pore space geometry and some assumptions related to physics are needed to derive the closed form analytical formulas. Such a comparison of the numerical results against an analytical solution has been performed by Alkhimenkov et al. (2020) for a different pore space geometry. Unfortunately, there is no analytical solution for the present study considering a periodic distribution of intersecting cracks in three-dimensions. But the qualitative comparison of the low and high-frequency limits (which correspond to relaxed and unrelaxed states) is possible (Mavko and Jizba, 1991). Several analytical studies show that that the anisotropy (described by Thompson's parameters) is, in general, more pronounced at high frequencies than at low frequencies (Guéguen, and Sarout, 2009, 2011). In the relaxed state, one can calculate the effective dry elastic moduli and use Gassmann's equations to obtain the effective moduli of the saturated medium. In the unrelaxed state, one can calculate the effective elastic moduli by restricting fluid flow (by using zero displacement boundary conditions in the cracks intersections). The low and high-frequency limits for elastic moduli have been calculated using these semi-analytical approaches and numerical results have been reproduced."

**Referee's comment 6**

In conclusion, this is an interesting paper.

25 January 2020 Yves Guéguen.

**Author's reply 6**

Thank you for your comments, which helped us to improve significantly the manuscript.

**Changes in the manuscript 6**

-

References:

Alkhimenkov, Y., Caspari, E., Gurevich, B., Barbosa, N. D., Glubokovskikh, S., Hunziker, J., & Quintal, B. (2020). Frequency-dependent attenuation and dispersion caused by squirt flow: Three-dimensional numerical study. Geophysics, 85(3), 1-71.

Bristow, J. R. (1960). Microcracks, and the static and dynamic elastic constants of annealed heavily cold-worked metals. British Journal of Applied Physics, 11,81–85.

Kachanov, M., & Mishakin, V. V. (2019). On crack density, crack porosity, and the possibility to interrelate them. International Journal of Engineering Science, 142, 185-189.

Guo, J., Rubino, J. G., Glubokovskikh, S., & Gurevich, B. (2017). Effects of fracture intersections on seismic dispersion: theoretical predictions versus numerical simulations. Geophysical Prospecting, 65(5), 1264-1276.

---

## Author Comment (AC2) · 20 Mar 2020

Referee's comment 1

I found the paper to be a good scientific contribution. I have recorded my comments in the attached pdf file. Please also note the supplement to this comment: https://www.solid-earth-discuss.net/se-2019-176/se-2019-176-RC1-supplement.pdf

Author's reply 1

Dear Reviewer,

Thank you very much for the time you have dedicated to review and comment our

manuscript. We believe that your comments have helped us to improve significantly the quality of the work. Please find below the responses to your comments.

Kind regards, Yury Alkhimenkov, on behalf of the authors

Changes in the manuscript 1

-

Referee's comment 2

In this work, the authors use numerical methods to study the azimuth, angle and frequency-dependent seismic velocities for cracked rocks due to squirt flow. The authors do a rigorous 3D numerical study using simple geometry and set up two different experiments – connected and disconnected cracks. The authors show that the seismic velocities are dependent on azimuth, angle and frequency only for connected cracks case. The authors also compare the results of the disconnected crack with the high frequency results of the connected cracks case. Apart from Thomsen's parameter that is primarily used for anisotropic studies in geophysics, the authors also formulate and use scalar parameter called anisotropy index parameter to describe the anisotropy dependence for squirt flow. This is a nice piece of numerical study for a simple geometry that gives new insights into anisotropy that arises due to squirt flow. The study is missing benchmarking of the numerical results with some simple analytical solutions or to some laboratory data. Benchmarking will help the readers have more confidence on the numerical results.

Author's reply 2

We compared the results of our numerical solver against an analytical solution for squirt flow in a recently published manuscript [Alkhimenkov et. al., 2020]. Thus, we confirm that our numerical results are correct and accurate. There is no analytical solution for the presented model geometry, only the low and high-frequency limits can be described analytically. We extended the discussion section to include a qualitative comparison to

the analytical solution.

Changes in the manuscript 2

Page 17, lines 352-365 (new subsection):

"A qualitative comparison against analytical models

Numerical simulations are useful but analytical models are especially attractive since they help us to better understand the physics and do not require sophisticated numerical simulations. The limitations of the analytical solutions are restricted to simple pore space geometry and some assumptions related to physics are needed to derive the closed form analytical formulas. Such a comparison of the numerical results against an analytical solution has been performed by Alkhimenkov et al. (2020) for a different pore space geometry. Unfortunately, there is no analytical solution for the present study considering a periodic distribution of intersecting cracks in three-dimensions. But the qualitative comparison of the low and high-frequency limits (which correspond to relaxed and unrelaxed states) is possible (Mavko and Jizba, 1991). Several analytical studies show that that the anisotropy (described by Thompson's parameters) is, in general, more pronounced at high frequencies than at low frequencies (Guéguen, and Sarout, 2009, 2011). In the relaxed state, one can calculate the effective dry elastic moduli and use Gassmann's equations to obtain the effective moduli of the saturated medium. In the unrelaxed state, one can calculate the effective elastic moduli by restricting fluid flow (by using zero displacement boundary conditions in the cracks intersections). The low and high-frequency limits for elastic moduli have been calculated using these semi-analytical approaches and numerical results have been reproduced."

Referee's comment 3

I believe that the authors could make some comments on the practical uses of this in some details apart from the introduction section of the paper where they mention a generic statement on it.

Author's reply 3

We have extended the introduction of the manuscript and included more details on the practical uses.

Changes in the manuscript 3

Page 1, lines 15-23: "Wave propagation is controlled by the effective rock properties. Wave velocity and attenuation can be estimated from seismic data in scenarios such as exploration seismic, seismology, borehole measurements and tomography. Rock physics could then be used to estimate different rock properties, such as mineral composition, elastic moduli, the presence of a fluid, pore space connectivity (and, hence, permeability) from seismic measurements. Thus, investigation of how cracks and fluids affect seismic properties has many practical applications. In activities including nuclear waste disposal, $CO_2$ geological sequestration, hydrocarbon exploration and production, geothermal energy production and seismotectonics, a quantification of the fluid content, porosity and permeability of rocks are of great interest. All these activities can benefit from rock physics studies, that is why cracked rocks have been under intensive studies during the last decades.

Referee's comment 4

Also, I think that the value of the work can be increased by application of the work on a real digital rock sample apart from the geometry that has been considered.

Author's reply 4

That's a very important point. A similar numerical workflow is used by my colleague Simon Lissa at the University of Lausanne to calculate the effective viscoelastic properties of 3D real digital rock samples. This project is ongoing.

Changes in the manuscript 4

-

Referee's comment 5

The authors can also think about presenting the figures for the results section in a better/simpler way (if possible). These figures are the crux of the paper and will be the most important thing for the readers to gain as a take-home message.

Author's reply 5

We agree that figures can be always improved, but at the same time, we don't want to simplify Figure 4 (for example) at the cost of showing less information. We insist on showing all components of the stiffness matrix in Figure 4, and later on simpler corresponding figures for P- and S-wave velocities are shown, as well as for the anisotropy measurements.

Changes in the manuscript 5

-

Referee's comment 6

I have mentioned my specific comments on the paper in the following section. General comments: Was any software (eg. COMSOL) used for the simulations?

Author's reply 6

Yes, we used a COMSOL Multiphysics PDE module where we implemented the equations in the weak form. Our results can be reproduced by using any open access finite element software. For that one needs a mesh generator (many open-access modules) and a solver of linear equations (many open access solvers).

Changes in the manuscript 6

Page 4, lines 98-100: "The COMSOL Multiphysics partial differential equation module is used for implementing equations (1) and (4) (displacement-stress formulation) in a weak form. Our numerical results can be fully reproduced by using any open-access

software which include mesh generation and finite-element implementation with a corresponding solver of a linear system of equations.

Referee's comment 7

Can there be a way to apply this work for a real 3D digital rock sample?

Author's reply 7

Yes, the implementation is straightforward. One needs to use a digital rock image for the geometry and, the most difficult, the cracks must be resolved. After segmentation, our numerical workflow can be used. My colleague Simon Lissa at the University of Lausanne is working on this right now.

Changes in the manuscript 7

-

Referee's comment 8

Can the results be benchmarked with some standard analytical solution or measurements from the laboratory?

Author's reply 8

We compared the results of our numerical solver against an analytical solution for squirt flow in a recently published manuscript [Alkhimenkov et. al., 2020]. Thus, we confirm that our numerical results are correct and accurate. There is no analytical solution for the presented model geometry. Only the low and high-frequency limits can be described analytically and it has been verified in our study. We extended the discussion section to include a qualitative comparison to the analytical solution.

Changes in the manuscript 8

Page 17 (new subsection), see also Changes in the manuscript 2

Referee's comment 9

Specific comments: Comments on Introduction section: 1 st line is a good motivation for the study but leaves the reader with the question why or how cracked rocks play a crucial role. Some references or a line to elaborate it would be effective. One might argue that the next one mentions hydraulic properties are affected by cracks is the justification, however, I still think it deserves some clarity.

Author's reply 9

We extended the introduction.

Changes in the manuscript 9

Page 1, lines 15-23: "Wave propagation is controlled by the effective rock properties. Wave velocity and attenuation can be estimated from seismic data in scenarios such as exploration seismic, seismology, borehole measurements and tomography. Rock physics could then be used to estimate different rock properties, such as mineral composition, elastic moduli, the presence of a fluid, pore space connectivity (and, hence, permeability) from seismic measurements. Thus, investigation of how cracks and fluids affect seismic properties has many practical applications. In activities including nuclear waste disposal, $CO_2$ geological sequestration, hydrocarbon exploration and production, geothermal energy production and seismotectonics, a quantification of the fluid content, porosity and permeability of rocks are of great interest. All these activities can benefit from rock physics studies, that is why cracked rocks have been under intensive studies during the last decades."

Referee's comment 10

Line 24 – Mesoscopic scale definition preferred before this line.Missing reference to Mavko and Nur (1975) classic paper that introduced "squirt flow"!

Author's reply 10

We rearranged the paragraph and put the mesoscopic scale definition in the correct order. We also added the reference.

Changes in the manuscript 10

page 2, line 44: "This phenomenon, known as squirt flow (Mavko and Nur, 1975) causes strong energy dissipation due to the viscosity of the fluid and the associated viscous friction."

Referee's comment 11

Line 51 – Das et al. (2019) Numerical simulation of coupled fluid-solid interaction at the pore scale: A digital rock-physics technology is the full paper in Geophysics and is a better reference than the conference abstract.

Author's reply 11

We changed the reference.

Changes in the manuscript 11

Page 2, line 57: "Das et al. (2019) numerically simulated a fully coupled fluid-solid interaction at the pore scale for digital rock samples."

Referee's comment 12

Comments on Numerical methodology section: Line 79 – Definition of u?

Author's reply 12

We added the definition.

Changes in the manuscript 12

Page 3, line 84: "u is the displacement vector"

Referee's comment 13

In the finite element numerical solver are you solving for displacement or velocity for the fluid phase (equation 3)? How is the coupling done in terms of displacement at the boundary between the two phases? It might be useful to mention a few lines here

instead of just references.

Author's reply 13

We are solving the equations using displacement-stress formulation in the solid and fluid subdomains. The same displacement unknowns describe the solid and fluid displacement in the different regions. The equations (2) and (3) are combined and written in such a way that no specific boundary condition between the two phases must be specified.

Changes in the manuscript 13

Page 4, line 89-95: "In the finite element numerical solver, equations 2-3 are combined in the space-frequency domain $\sigma_{ij} = \lambda e \delta_{ij} + 2\mu\epsilon_{ij} + i\omega \left(2\eta\epsilon_{ij} - \frac{2}{3}\eta e\delta_{ij}\right), where \epsilon_{ij}$ are the components of the strain tensor $\epsilon_{ij} = 0.5\left(u_{i,j} + u_{j,i}\right)$, $e$ is the trace of the strain tensor, $\lambda$ and $\mu$ are the Lame parameters, $u_i$ is the displacement in the $i$-th direction, $\delta_{ij}$ is the Kronekecker delta, $i$ is the imaginary unit and $\omega$ is the angular frequency. In the domain representing a solid material, the equation 4 reduces to equation 2 by setting the shear viscosity $\eta$ to zero. In the domain representing compressible viscous fluid, equation 3 is recovered by setting the shear modulus $\mu$ to zero."

Page 4, line 100: "The COMSOL Multiphysics partial differential equation module is used for implementing equations (1) and (4) (displacement-stress formulation) in a weak form."

Referee's comment 14

Line 86 – Why are the inertial terms neglected? What would be the limitations that might arise due to this approximation? Any discussion on how this can be included?

Author's reply 14

Inertial terms are neglected. When a propagating wave has a wavelength much larger than the inhomogeneities inside the medium, then the inhomogeneous medium prop-

erties can be replaced by its averaged properties, which correspond to the quasistatic experiment. The main limitation is the ratio between the wavelength $\lambda$ and the length of our model domain L, so our results are valid when L divided by $\lambda <<1$. When this condition is not satisfied, then inertial terms are important and some wave scattering will be present. The inertial terms can be included but it will not affect our study since we are in the regime where inertial effects are negligible. The present author is working on a different study in the framework of poroelasticity considering inertial effects, so these results will be available in the future.

Changes in the manuscript 14

-

Referee's comment 15

Line 88 – Why is the PARDISO solver used? The reader might wander what are the advantages of using this versus any other solver.

Author's reply 15

We tested several available solvers and found that PARDISO solver gives the best performance for our numerical model configuration and computing system. MUMPS (MUltifrontal Massively Parallel Sparse direct Solver) converges to the same result as PARDISO but it uses 30% more RAM memory and two times slower. Iterative solvers could not converge. That is most likely due to a very heavy numerical model. We believe that PARDISO solver is good only for our specific applications, therefore we do not discuss this choice. For other model geometry, another solver might give better results. For example, a colleague of mine who is working on 3D digital rock images found that MUMPS direct solver is more robust than PARDISO solver.

Changes in the manuscript 15

-
[Figure]

Referee's comment 16

Line 89 – What is direct relaxation test? Reference/ one line description can be useful.

Author's reply 16

A direct relaxation test is a numerical test when we apply specific displacement boundary conditions in such a way that one specific component of the stiffness matrix can be calculated. So the word "direct" has the meaning that one specific cij component is calculated at the end. (A more detailed explanation of boundary conditions is presented in the appendix A.) This "direct approach" is in contrast to approaches that perform, for example, 4 relaxation tests to calculate 5 stiffness components, one or two "indirectly".

Changes in the manuscript 16

Page 4, lines 103-107: "The basic idea of the direct relaxation tests is that a displacement boundary condition of the form $u = 10^{-6} \times \exp(i\omega t)$ is applied to a certain external wall of the model and in a certain direction, while at other walls of the model, the displacements are set to zero or let free to change. In the direct tests that we perform, only one component of the stiffness matrix cij can be directly calculated after one numerical simulation. A detailed description of the boundary conditions is given in Alkhimenkov et al. (2020)."

Referee's comment 17

Line 96 – Is the harmonic displacement a function of time? Are you solving the equations in time domain or frequency domain? How does combining equations 2 and 3 work if the equations are solved in time domain vs frequency domain?

Author's reply 17

In our numerical simulations, the harmonic displacement is a function of frequency and we solve equations in the space-frequency domain. In other words, time derivatives are replaced by $i\omega$. The coupling is the same in the time domain and in the frequency

domain, the only difference is $i\omega$. We performed our study in the frequency domain, therefore, we do not discuss how to solve the corresponding equations in the time domain.

Changes in the manuscript 17

Page 4, lines 88-96: "In the finite element numerical solver, equations (2)-(3) are combined in the space-frequency domain (see equation 4).

Also, see  Changes in the manuscript 16"

Referee's comment 18

Line 100 – What are mixed direct tests?

Author's reply 18

This is explained in Appendix A. A mixed direct test is similar to the direct relaxation test. Diagonal components of the stiffness matrix cij can be calculated directly, i.e. by using direct tests. A mixed test for non-diagonal components (c12, c13, c23) is necessary plus the result of the direct test of the corresponding diagonal element to calculate the non-diagonal elements (c12, c13, c23). In Appendix A we presented the corresponding equations as well as boundary conditions for direct and mixed direct tests. Also, see  Changes in the manuscript 16.

Changes in the manuscript 18

-

Referee's comment 19

Comments on Numerical model: Line 108 Any specific reason for having an aspect ratio of 0.01?

Author's reply 19

There is always a compromise between the crack aspect ratio and the possibility to

perform the numerical simulation. In our simulations, we need several elements inside the crack to resolve fluid flow and the parabolic fluid velocity profile. Therefore, if the aspect ratio is very low (meaning that for a given crack length one decreases crack thickness), one will need too many elements, so at some point, it becomes impossible to solve the problem due to RAM memory limitations. Therefore, we found a compromise. With such an aspect ratio of 0.01, we have enough elements inside the crack and the total number of elements is not too big (around 3 million), so we can solve the problem.

Changes in the manuscript 19

-

Referee's comment 20

Line 109 Any specific reason for using glycerol?

Author's reply 20

Yes. In the laboratory, glycerol is usually used in order to shift the characteristic frequency to lower frequencies and, thus, to measure dispersion and attenuation. We used glycerol also for this purpose, and consequently to qualitatively compare our characteristic frequency to that of a forced oscillation laboratory experiment.

Changes in the manuscript 20

-

Referee's comment 21

Line 110 In Table 1, how is the Bulk Modulus (K) of the fluids used in equation 3?

Author's reply 21

There is a fluid pressure p in equation 3. The pressure is the volumetric part of the stress tensor with a "minus" sign. The volumetric part of the stress tensor is related to

the strain tensor via the fluid bulk modulus K.

Changes in the manuscript 21

See  Changes in the manuscript 13

Referee's comment 22

Line 111 Can there be some quantitative ways of describing fine and coarser mesh – size of the elements (maximum and minimum size, average size of the mesh element) including the details mentioned in the caption of Figure 2.  Also, along the z axis of the mesh, parts of it has 3 elements as per the snapshot in Figure 2. Is this sufficient number of elements from numerical point of view?  Is there any way to justify that the numerical solution is stable with the number of elements used?

Author's reply 22

We have the mesh sizes (minimum and maximum) in the caption to figure 2.  The numerical tests suggest that 3 elements along the z-axis is a sufficient number of elements to converge to the correct result, we performed several tests similar to those of Quintal et. al, (2019).  The stability can be justified in a naive way, by increasing the number of elements and comparing solutions, which we also tested.

Changes in the manuscript 22

-

Referee's comment 23

Line 113 What type of system configuration was used for making the simulations?  It seems like 0.95TB of RAM would require very specific type of machine to run.  How much time did it take to run each of the simulation?

Author's reply 23

We used an intel node: dual socket E5-2683 v4 @ 2.1GHz (32 cores), 1024 GB RAM.

For each frequency, it takes about 2.5 hours. There are 12 calculations for each component, 9 components of the stiffness matrix were calculated. So, for each model, it took 2.5x12x9=270 hours to calculate the full stiffness matrix. Since we have two models (with connected and disconnected cracks), the total number is 540 computer hours giving 22.5 days. In total more calculations were performed because we checked convergence and performed other tests, so in the end, the total number of machine-hours is even larger.

Changes in the manuscript 23

Page 5, lines 129-132:

"The simulation is performed for 12 different frequencies from $10^1$ to $10^{6.5}$ Hz for each of the nine components of the stiffness matrix (c11, c22, c33, c12, c13, c23, c44, c55, c66). For each frequency, the solver uses approximately 0.95 Terabyte of RAM memory and takes approximately 2.5 hours on 32 Intel dual-socket E5-2683 v4 2.1 GHz (1024 GB RAM) cores."

Referee's comment 24

Line 126-127 Does the difference between the stiffness of the two disconnected cracks also depend on the boundary conditions applied to the system? In other words, will the stiffness be always different irrespective of the boundary condition applied?

Author's reply 24

Our results for the effective properties are independent of our boundary conditions. However, several studies show that, for example, stress and strain boundary conditions might give different results and this has to be carefully considered (Milani et al., 2016, Geophysics). With respect to the difference between connected and disconnected cracks, there will be always a difference present for any type of boundary condition.

Changes in the manuscript 24

-

Referee's comment 25

Comments on Results section:Line 147 Unclear what is meant by the cracks can be described by only two compliances as per equation 6?

Author's reply 25

The effect of a crack embedded into the linear elastic material can be quantified by the means of the so-called stiffness contribution tensor or alternatively by the two parameters — normal and tangential crack compliances $Z_n$ and $Z_t$. For that, we need an effective compliance matrix of the background material and crack compliance $Z_n$ and $Z_t$. It can be seen from equation 7: once we invert the matrix $c_{ij}$ and get the compliance matrix $s_{ij}$, $Z_n$ and $Z_t$ can be inverted ($Z_n$ can be inverted from $s_{22}$ or $s_{33}$ and $Z_t$ can be inverted from $s_{44}$).

Changes in the manuscript 25

-

Referee's comment 26

I am also confused that in Line 142 it is mentioned that there is a significant difference due to vertical crack separation. However, in Line 148, it is mentioned that only connected cracks case is considered? Can this be explained better?

Author's reply 26

We removed this sentence which is indeed confusing.

Changes in the manuscript 26

page 8. Lines 164-170, removed one sentence

Referee's comment 27

Line 165 – What are the values of frequencies used for LF, Fc and HF?

Author's reply 27

The values are $10^1$ Hz for LF, $10^4$ Hz for Fc and $10^{6.5}$ HZ for HF.

Changes in the manuscript 27

We modified the caption of figure 3.

"Snapshots of the fluid pressure Pf in the cracks at three different frequencies: LF - the low-frequency limit (corresponds to $10^1$ Hz, relaxed state), Fc - intermediate frequency snapshot (corresponds to $10^4$ Hz, close to the characteristic frequency) and HF - the high-frequency limit (corresponds to $10^{6.5}$ Hz, unrelaxed state)."

Referee's comment 28

Line 173 – It is unclear what is meant by the elastic limit in this case? Can this be compared to any standard elastic limits that are mentioned in literature?

Author's reply 28

In our numerical simulation, the grain material is elastic while fluid is not, fluid can diffuse which is described by viscosity. In the high-frequency limit, there is no time for fluid to flow, so it behaves as an elastic material having bulk properties of glycerol and zero shear modulus. We do not understand exactly what the referee meant by the standard elastic limit but the configuration simply corresponds to a fully elastic material where the fluid behaves as a solid with zero shear modulus.

Changes in the manuscript 28

-

Referee's comment 29

Figure 4 might be improved from a reader's perspective.

[Figure]

Author's reply 29

Simplified information derived from the data presented in Figure 4 is shown in the subsequent figures. We know that it shows a lot of curves, but we prefer showing everything in this figure for the sake of completeness.

Changes in the manuscript 29

-

Referee's comment 30

Line 180 – Even in figure 4a, the Real part of the c22 and c33 component coincides. Is there a specific reason for pointing out separately the attenuation and dispersion components are the same due to symmetry of the model?

Author's reply 30

The dispersion and attenuation of c22 and c33 components are the same due to the symmetry of the model and the same geometrical characteristics of the cracks. For example, the model can be symmetric but the two cracks might have different stiffness due to different fluid or asperities of the crack walls, etc.

Changes in the manuscript 30

-

Referee's comment 31

Line 183 – It might be useful to explain what negative attenuation means in physical sense. Is there a reason for the negative attenuation behavior?

Author's reply 31

Now, this is explained in the text.

Changes in the manuscript 31

Page 4, lines 118-120"

"Note that usually the inverse quality factor is used as a measure of atten-uation (O'connell and Budiansky, 1978). In this study, we show the inverse quality factor for each component of the stiffness tensor, even though the ratio Im(cij(omega))=Re(cij(omega)) does not represent attenuation of any corresponding wave mode for some components."

Page 9, Lines 199-209: "The c12 and c13 components are non-dispersive, the c23 component exhibits strong negative dispersion and a negative inverse quality factor peak shifted towards high frequencies compared to that of the c22, c33 components. A similar phenomenon has been reported by Guo et al. (2017) in the context of two-dimensional simulations. The c23 component does not correspond to a wave mode alone, it is always used together with c22 or/and c33 components. Therefore, no wave will gain energy. This negative inverse quality factor sign for the c23 component was also verified using Kramers-Kronig relations. In other words, different components of the stiffness tensor might have positive or negative values of the ratio Im(c23)=Re(c23) but when we calculate the velocity and the inverse quality factor of a wave, the cumu-lative effect of all cij components must be physical and no negative attenuation will be observed."

Referee's comment 32

Line 194 – It might be useful to mention how the seismic velocities are calculated as a function of phase?

Author's reply 32

Now, this is better explained in the text.

Changes in the manuscript 32

page 10, lines 118-220: "The P- and S-wave phase velocities are calculated by solving the Christoffel equation which represents an eigenvalue problem relating the stiffness

components cij , the phase velocities of plane waves that propagate in the medium and the polarization of the waves (Fedorov, 1968; 220 Tsvankin, 2012)."

Referee's comment 33

Figure 6 – Is there a reason for the discrepancy between $V_{SV}$ between the disconnected crack model and the high-frequency result at a phase angle of 0, 90 and 180 degrees?

Author's reply 33

Yes, indeed, there is a slight discrepancy between the connected and disconnected crack models in terms of $V_{SV}$ velocities at specific angles. In fact, such discrepancy is very small, less than 0.6 %. The nature of this discrepancy is due to the crack separation. If the model is dry, the discrepancy is huge, as explained in the paper. For the fluid-saturated model, the discrepancy is small but it still exists and can be visible at $V_{SV}$ velocities.

Changes in the manuscript 33

page 11, lines 236-238: "A slight discrepancy (around 0.5%) between the SV-wave velocities for the disconnected crack model (Figure 6, dashed red line) and the high-frequency velocity for the connected crack model (Figure 6, green line) at phase angles of 0, 90 and 180 degrees is due to the crack separation."

Referee's comment 34

Comments related to Discussions: Line 311 – What are the model parameters that the conclusion will depend upon? The authors mention about the need of sensitivity analysis of the model parameters. It would be useful to give a qualitative idea about the possible model parameters that the results will depend on.

Author's reply 34

That is an important point and the most difficult to answer. Basically, all model param-

eters affect the effective response. In order to give a precise answer one must run the sensitivity study for all parameters. Since for one set of parameters we needed around 22 days of continuous calculations, for a decent sensitivity study, it would require years with our current computational power. In our opinion, the most important parameters are the crack density, crack compliance, crack orientation and the saturation of the cracks.

Changes in the manuscript 34

Page 18, lines 380-387: "In summary, squirt flow does affect effective mechanical properties of cracked rocks and, thus, seismic velocity anisotropy. Given that seismic anisotropy variations with frequency are very sensitive to the pore space geometry and material properties, it is difficult to make a general prediction. According to our study, the effective frequency-dependent response of a cracked medium is different in different planes. The local response (in a certain plane) is controlled by cracks orientation, which is the key parameter. The magnitude of the frequency-dependent response (i.e. the dispersion and attenuation) is controlled by crack compliances, crack porosity and their fluid content (dry or liquid-saturation will cause completely different behavior). Most importantly, crack porosity is a very important parameter in fluid-saturated rocks (contrary to dry rocks) since it defines the volume of fluid content which may flow due to wave propagation, causing wave attenuation and dispersion."

References:

Alkhimenkov, Y., Caspari, E., Gurevich, B., Barbosa, N. D., Glubokovskikh, S., Hunziker, J., Quintal, B. (2020). Frequency-dependent attenuation and dispersion caused by squirt flow: Three-dimensional numerical study. Geophysics, 85(3), 1-71.

Quintal, B., Caspari, E., Holliger, K., Steeb, H. (2019). Numerically quantifying energy loss caused by squirt flow. Geophysical Prospecting, 67(8), 2196-2212.

Milani, M., Rubino, J. G., Müller, T. M., Quintal, B., Caspari, E., Holliger, K. (2016).

[Figure]

Representative elementary volumes for evaluating effective seismic properties of heterogeneous poroelastic mediaREVs for heterogeneous porous media. Geophysics, 81(2), D169-D181.